# Sleep deprivation causes memory deficits by negatively impacting neuronal connectivity in hippocampal area CA1

**Robbert Havekes[1,2]\*, Alan J Park[1]†, Jennifer C Tudor[1], Vincent G Luczak[1], Rolf T Hansen[1], Sarah L Ferri[1], Vibeke M Bruinenberg[2], Shane G Poplawski[1], Jonathan P Day[3], Sara J Aton[4], Kasia Radwańska[5], Peter Meerlo[2], Miles D Houslay[6], George S Baillie[3], Ted Abel[1]\***

[1]Department of Biology, University of Pennsylvania, Philadelphia, United States; [2]Groningen Institute for Evolutionary Life Sciences (GELIFES), University of Groningen, Groningen, The Netherlands; [3]Institute of Cardiovascular and Medical Science, College of Medical, Veterinary and Life Sciences, University of Glasgow, Glasgow, United Kingdom; [4]LSA Molecular, Cellular, and Developmental Biology, University of Michigan-Ann Arbor, Ann Arbor, United States; [5]Laboratory of Molecular Basis of Behavior, Head Nencki Institute of Experimental Biology, Warsaw, Poland; [6]Institute of Pharmaceutical Science, King's College London, London, United Kingdom

**\*For correspondence:** r.havekes@rug.nl (RH); abele@sas.upenn.edu (TA)

**Present address:** †Department of Psychiatry Columbia University, New York State Psychiatric Institute, New York, United States

**Competing interests:** The authors declare that no competing interests exist.

**Abstract** Brief periods of sleep loss have long-lasting consequences such as impaired memory consolidation. Structural changes in synaptic connectivity have been proposed as a substrate of memory storage. Here, we examine the impact of brief periods of sleep deprivation on dendritic structure. In mice, we find that five hours of sleep deprivation decreases dendritic spine numbers selectively in hippocampal area CA1 and increased activity of the filamentous actin severing protein cofilin. Recovery sleep normalizes these structural alterations. Suppression of cofilin function prevents spine loss, deficits in hippocampal synaptic plasticity, and impairments in long-term memory caused by sleep deprivation. The elevated cofilin activity is caused by cAMP-degrading phosphodiesterase-4A5 (PDE4A5), which hampers cAMP-PKA-LIMK signaling. Attenuating PDE4A5 function prevents changes in cAMP-PKA-LIMK-cofilin signaling and cognitive deficits associated with sleep deprivation. Our work demonstrates the necessity of an intact cAMP-PDE4-PKA-LIMK-cofilin activation-signaling pathway for sleep deprivation-induced memory disruption and reduction in hippocampal spine density.

## Introduction

Sleep is a ubiquitous phenomenon and most species, including humans, spend a significant time asleep. Although the function of sleep remains unknown, it is widely acknowledged that sleep is crucial for proper brain function. Indeed, learning and memory, particularly those types mediated by the hippocampus, are promoted by sleep and disrupted by sleep deprivation (*Havekes et al., 2012a*; *Abel et al., 2013*; *Whitney and Hinson, 2010*). Despite the general consensus that sleep deprivation impairs hippocampal function, the molecular signaling complexes and cellular circuits by which sleep deprivation leads to cognitive deficits remain to be defined.

The alternation of wakefulness and sleep has a profound impact on synaptic function, with changes observed in synaptic plasticity and transmission (*Havekes et al., 2012a*; *Abel et al., 2013*;

**eLife digest** The demands of modern society means that millions of people do not get sufficient sleep on a daily basis. Sleep deprivation, even if only for brief periods, can impair learning and memory. In many cases, this impairment appears to be related to changes in the activity of a brain region called the hippocampus. However, the exact processes responsible for producing the effects of sleep deprivation remain unclear.

During learning or forming a new memory, the connections between the relevant neurons in the brain change. Havekes et al. found that depriving mice of sleep for just five hours dramatically reduced the connectivity between neurons in the hippocampus. This reduction is caused by the increased activity of cofilin, a protein that breaks down the actin filaments that shape the connections between neurons.

Havekes et al. then used a virus to introduce an inactive version of cofilin into hippocampal neurons to suppress the activity of the naturally present cofilin. This manipulation prevented both the loss of the connections between neurons and the memory deficits normally associated with sleep deprivation. Havekes et al. also found that recovery sleep leads to the re-wiring of neurons in the hippocampus. Future studies are now needed to determine how the neurons are able to re-wire themselves during recovery sleep.

*Tononi and Cirelli, 2014*). This relationship has led to the development of influential theories on the function of sleep (*Tononi and Cirelli, 2006*; *Pavlides and Winson, 1989*). Recent imaging suggests that dendritic structure is dynamic, especially during development, with alterations in spine numbers correlating with changes in sleep/wake state (*Maret et al., 2011*; *Yang and Gan, 2012*). However, the impact of sleep deprivation or sleep on synaptic structure in the hippocampus in the context of memory storage or synaptic plasticity has not been examined. This is an important issue, as such structural changes in ensembles of synapses have been shown to play a critical role in memory storage (*Caroni et al., 2012*; *Vogel-Ciernia et al., 2013*).

The formation of associative memories increases the number of dendritic spines in area CA1 of the hippocampus (*Leuner et al., 2003*). Also, the induction of long-term potentiation (LTP), a cellular correlate of memory storage (*Mayford et al., 2012*), is associated with an increase in spine density in cultured hippocampal neurons (*Oe et al., 2013*). In addition to a critical function during development (*Gurniak et al., 2005*), cofilin plays an essential role in synapse structure by mediating both the enlargement and pruning of dendritic spines (*Rust, 2015*; *Bamburg, 1999*; *Bosch et al., 2014*). The activity of cofilin is negatively regulated by phosphorylation. Specifically, phosphorylation of serine 3 of cofilin suppresses its depolymerizing and F-actin severing activity (*Bamburg, 1999*). Importantly, increased cofilin activity can lead to the depolymerization and severing of F-actin, which in turn results in the shrinkage and loss of spines (*Rust, 2015*; *Zhou et al., 2004*; *Davis et al., 2011*; *Shankar et al., 2007*). Hippocampal cofilin phosphorylation levels are increased after the induction of long-term potentiation (LTP) (*Rex et al., 2010*; *Chen et al., 2007*; *Briz et al., 2015*), and during memory consolidation (*Fedulov et al., 2007*; *Suzuki et al., 2011*). Additionally, elevated cofilin activity in the hippocampus was recently implicated in abnormal spine structure and function in mutant mice with altered chromatin remodeling (*Vogel-Ciernia et al., 2013*).

Here we show for the first time that 5 hr of sleep deprivation leads to the loss of dendritic spines of CA1, but not CA3, neurons in the dorsal hippocampus. The spine loss in CA1 neurons was accompanied by reductions in dendrite length. This process was readily reversed by sleep, with just 3 hr of recovery sleep normalizing this spine loss and dendrite length. The molecular mechanisms underlying these negative effects of sleep deprivation were shown to target cofilin, whose elevated activity could contribute to spine loss. Indeed, suppression of cofilin activity in hippocampal neurons prevented the structural, biochemical, and electrophysiological changes as well as the cognitive impairments associated with sleep loss. The elevated cofilin activity is caused by the activity of the cAMP degrading phosphodiesterase-4A5 isoform (PDE4A5), which suppresses activity of the cAMP-PKA-LIMK pathway. Genetic inhibition of the PDE4A5 isoform in hippocampal neurons restores LIMK and cofilin phosphorylation levels and prevents the cognitive impairments associated with sleep loss.

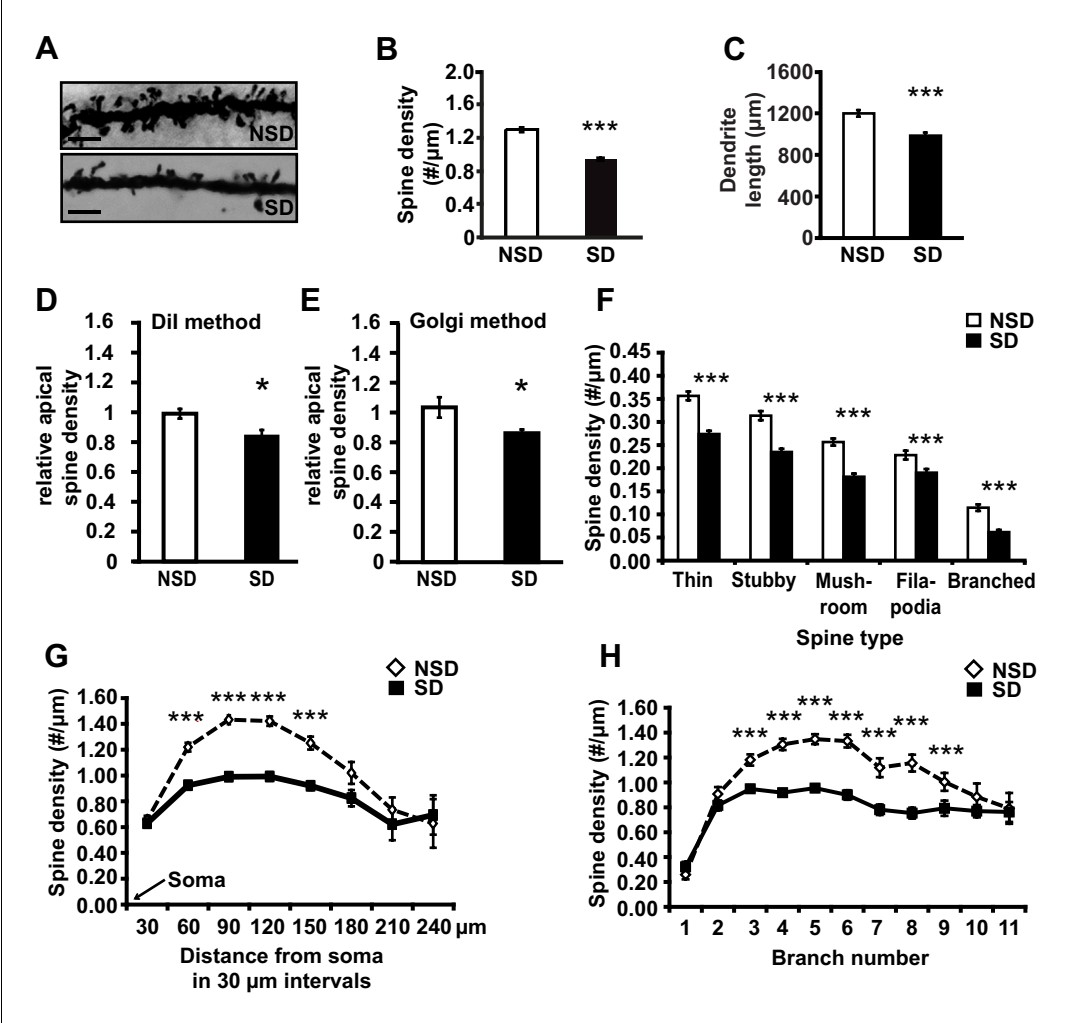

**Figure 1.** Sleep deprivation reduces spine numbers and dendrite length in CA1 neurons of the hippocampus. (A) Representative images of Golgi-impregnated dendritic spines of CA1 pyramidal neurons from sleep deprived (SD) and non-sleep deprived (NSD) mice. Scale bar, 5 μm. (B) Sleep deprivation reduces the spine density of apical/basal dendrites of CA1 neurons (n = 5–6, Student's t-test, p=0.0002). (C) Sleep deprivation decreases apical/basal dendrite length of CA1 neurons (n = 5–6, Student's t-test, p=0.0012). (D, E) Comparative analyses of spine numbers in the second-third branch of apical dendrites of CA1 neurons reveal a significant reduction as a result of sleep deprivation using either the DiI labeling method (n = 3–4, Student's t-test, p=0.03) or Golgi analyses (n = 5, Student's t-test, p=0.03). Importantly, for the comparison of the two methods we focused on the second and third branch of the apical dendrites. See also the Materials and methods section. (F) Sleep deprivation reduces the number of all spine types in apical/basal dendrites of CA1 neurons (n = 5–6, Student's t-test, p<0.005). (G) Sleep deprivation reduces spine density of apical/basal dendrites between 60 and 150 μm away from the soma of CA1 neurons (n = 5–6, Student's t-test, p<0.005). (H) Sleep deprivation reduces apical/basal spine density in branch 3–9 of CA1 neurons (n = 5–6, Student's t-test, p<0.005). NSD: non-sleep deprived, SD: sleep deprived, Values represent the mean ± SEM. *p<0.05, ***p<0.005, by Student's t test. See also *Figure 1—figure supplement 1* and *2* for separate Golgi analyses of apical and basal spine numbers.

The following figure supplements are available for figure 1:

**Figure supplement 1.** Sleep deprivation decreases spine density and dendrite length in both basal and apical dendrites of CA1 neurons.

**Figure supplement 2.** Sleep deprivation does not reduce spine density and dendrite length in both basal and apical dendrites of CA3 neurons.

Thus changes in the cAMP-PDE4-PKA-LIMK-cofilin signaling pathway in the adult hippocampus underlie the cognitive deficits associated with sleep loss. These observations provide a molecular model for the notion that prolonged wakefulness reduces structural signaling and negatively impacts dendritic structure, which is then restored with sleep.

## Results

### Sleep deprivation causes a robust reduction in apical and basal CA1 spine numbers and dendrite length

To determine whether short periods of sleep loss affect dendritic structure in the hippocampus, we used Golgi staining to examine the length of dendrites and number of dendritic spines in the mouse hippocampus following 5 hr of sleep deprivation, a period of sleep loss that is known to impair selectively hippocampus-dependent memory consolidation and synaptic plasticity (*Havekes et al., 2012a*; *Abel et al., 2013*; *Graves et al., 2003*; *Vecsey et al., 2009*; *Havekes et al., 2014*). Analyses of Golgi-impregnated CA1 neurons (*Figure 1A*) indicated that sleep deprivation significantly reduced the apical/basal spine density (*Figure 1B*; spine numbers per dendrite, NSD: 1.42 ± 0.03, SD: 1.17 ± 0.02; Student's t-test, p=0.0002) and dendrite length (*Figure 1C*; NSD: 1198.4 ± 31.6, SD: 984.2 ± 29.8 μm; Student's t-test, p=0.0012). This decrease in spine density and dendrite length was observed in both apical and basal dendrites (*Figure 1—figure supplement 1A,B*). To complement our Golgi studies, we conducted an additional experiment in which individual CA1 neurons in hippocampal slices from sleep deprived and non-sleep deprived mice were labeled using the DiI method as described (*Seabold et al., 2010*). In line with our Golgi studies, we found that sleep deprivation significantly reduced the total number of spines on apical dendrites of CA1 neurons (*Figure 1D*; NSD: 1.0 ± 0.03, SD: 0.84 ± 0.04 Student's t-test, p=0.033; *Figure 1E*; NSD: 1.0 ± 0.06, SD: 0.86 ± 0.02 Student's t-test, p=0.03).

Subtype-specific apical/basal spine analyses of the Golgi impregnated neurons revealed a significant decrease for all spine subtypes in sleep-deprived mice (*Figure 1F*, for all spine types, Student's t-tests p<0.005, for separate apical and basal spine analyses see Supplementary *Figure 1C,D*). Sleep deprivation causes the greatest reduction in apical/basal spine density between 60 μm and 150 μm from the soma (*Figure 1G*, for separate apical and basal spine analyses see *Figure 1—figure supplement 1E,F*). This region corresponds to the middle range of the dendritic branch (third to ninth branch orders, *Figure 1H*) where the primary input from CA3 is located (*Neves et al., 2008*), suggesting that the hippocampal Schaffer collateral pathway is particularly vulnerable to sleep loss.

We next assessed whether sleep deprivation also impacted spine numbers and dendrite length of CA3 neurons. Surprisingly, in contrast to CA1 neurons, CA3 neurons were unaffected by sleep deprivation. We did not observe reductions in spine density or dendrite length of either basal or apical dendrites of any type (*Figure 1—figure supplement 2*). Together, these data suggest that CA1 neurons at the level of dendritic structure seem particularly vulnerable to sleep deprivation.

To determine whether recovery sleep would reverse spine loss in CA1 neurons, we repeated the sleep deprivation experiment but then left the sleep-deprived mice undisturbed for three hours afterwards. This period was chosen as our previous work indicated that three hours of recovery sleep is sufficient to restore deficits in LTP caused by sleep deprivation (*Vecsey et al., 2009*). In line with the electrophysiological studies, recovery sleep restored apical/basal spine numbers and dendrite length in CA1 neurons to those observed in non-sleep deprived mice (*Figure 2A*, spine density of apical/basal dendrites, NSD: 1.23 ± 0.02, RS: 1.29 ± 0.02; Student's t-test, p>0.05; *Figure 2B*, dendrite length in μm, NSD: 1817.0 ± 64.6, RS: 1741.6 ± 55.57; Student's t-test, p=0.1721; *Figure 2C*, Student's t-test, p>0.05 for each distance from soma; *Figure 2D*, Student's t-test, p>0.05 for each branch number; for separate apical and basal spine analyses see *Figure 2—figure supplement 1*) with the exception of branched spines in the basal dendrites (*Figure 2—figure supplement 1C*). Recovery sleep slightly but significantly elevated the number of filopodia spines of the apical CA1 dendrites and total spine numbers of the seventh and eighth branch of the apical and basal dendrites respectively (*Figure 2—figure supplement 1*).

### Sleep deprivation increases hippocampal cofilin activity and suppression of cofilin function prevents spine loss in CA1 neurons associated with the loss of sleep

We hypothesized that the structural changes in the hippocampus following sleep deprivation might be related to increased activity of the actin-binding protein cofilin because increased cofilin activity can cause shrinkage and loss of dendritic spines through the depolymerization and severing of actin filaments (*Zhou et al., 2004*; *Davis et al., 2011*; *Shankar et al., 2007*). The ability of cofilin to bind

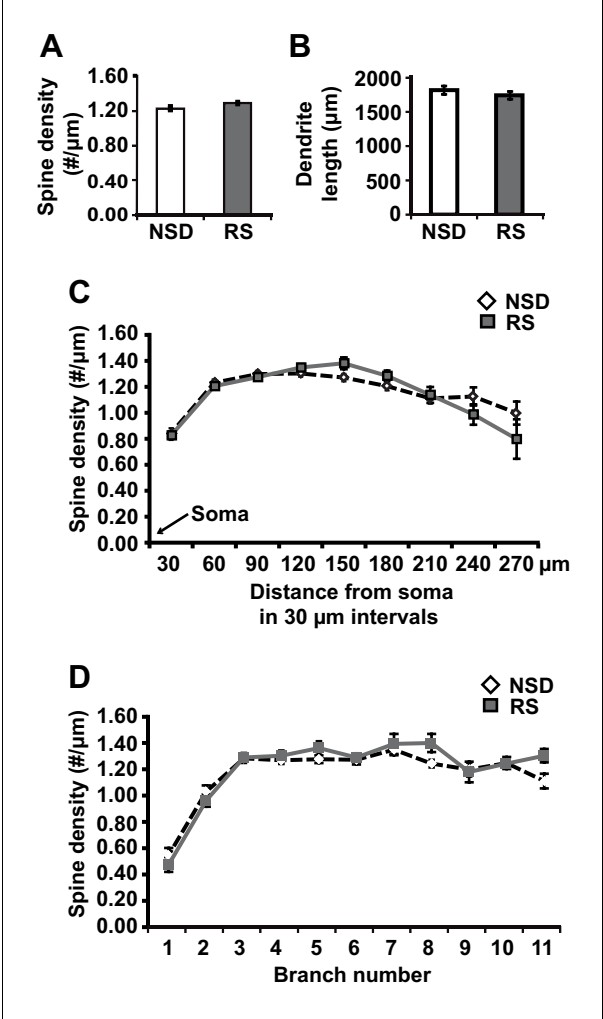

**Figure 2.** Three hours of recovery sleep restores spine numbers and dendrite length of CA1 neurons in the hippocampus. (**A**) Golgi analyses indicated that three hours of recovery sleep after 5 hr of sleep deprivation restores the total number of spines per apical/basal dendrite of CA1 neurons (n = 6, Student's t-test, p>0.05). (**B**) Three hours of recovery sleep after 5 hr of sleep deprivation restores apical/basal dendrite length of CA1 neurons (n = 6, Student's t-test, p=0.173). (**C, D**) Three hours of recovery sleep restores apical/basal spine numbers at all distances from the soma (Student's t-test, p>0.05 for each distance from soma, **C**) and at each branch number (Student's t-test, p>0.05 for each branch number, **C**). NSD: non-sleep deprived, RS: Sleep deprivation + recovery sleep. Values represent the mean ± SEM. See also *Figure 2—figure supplement 1* for separate Golgi analyses of apical and basal spine numbers.

The following figure supplement is available for figure 2:

**Figure supplement 1.** Three hours of recovery sleep after 5 hr of sleep deprivation is sufficient to restore spine numbers and dendrite length in both basal and apical dendrites of CA1 neurons.

---

and depolymerize and sever F-actin is inhibited by phosphorylation at serine 3 (*Rust, 2015*; *Bamburg, 1999*; *Bosch et al., 2014*). We therefore assessed whether sleep deprivation alters cofilin phosphorylation by Western blot analysis of hippocampus homogenates collected after 5 hr of sleep deprivation. Indeed, 5h of sleep deprivation reduced cofilin Ser-3 phosphorylation, suggesting an increase in cofilin activity in the hippocampus (NSD: 100.0 ± 6.9%; SD: 67.7 ± 9.2%; Student t-test p=0.0090; *Figure 3A*). A similar effect was not evident in the prefrontal cortex (NSD, n = 5: 100.0 ± 1.84%; SD, n = 5: 101.92 ± 2.41%; Student t-test p=0.54; *Figure 3—figure supplement 1*), indicating sleep deprivation affects cofilin phosphorylation in a brain region-specific fashion.

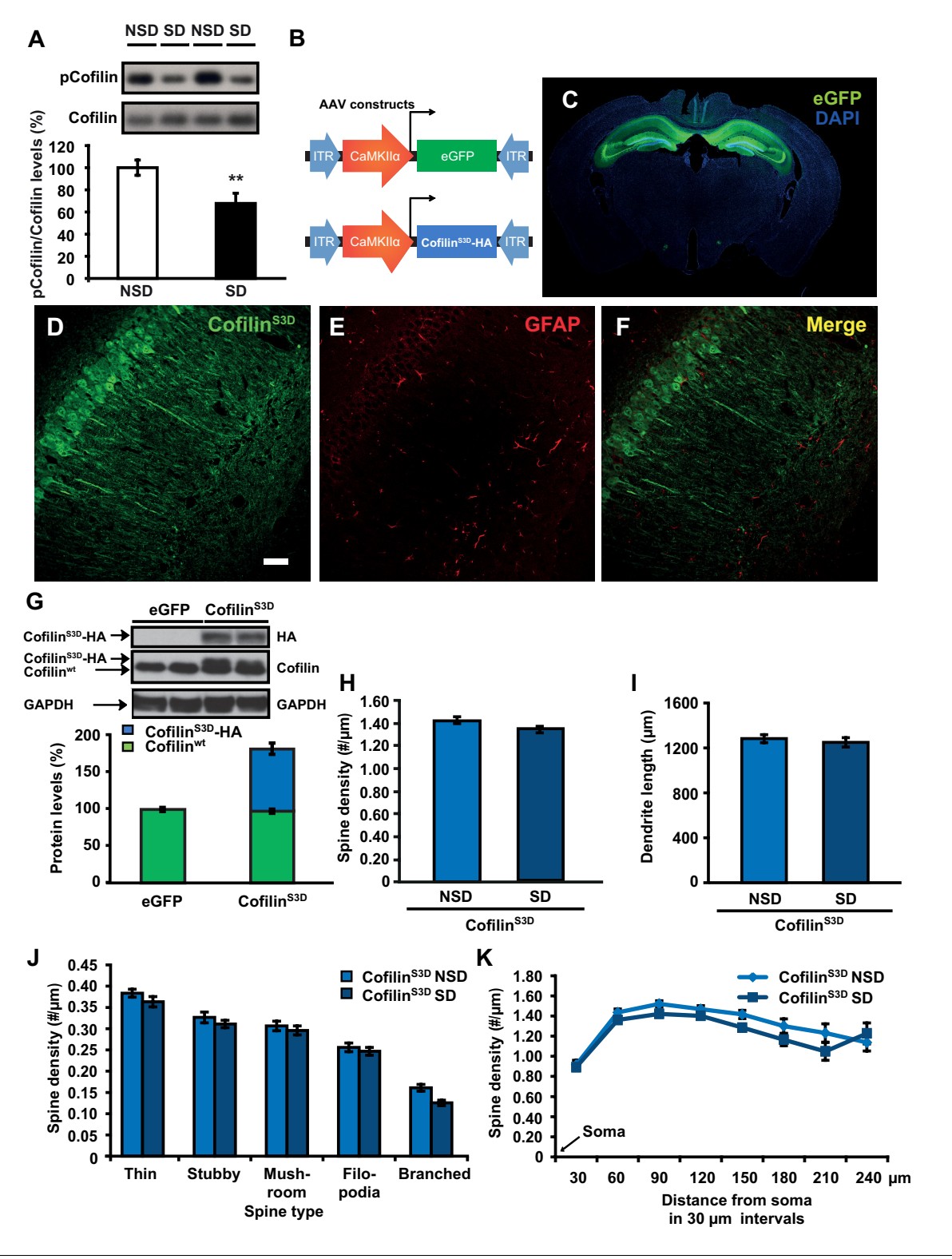

**Figure 3.** Increased cofilin activity in the hippocampus mediates the spine loss associated with sleep deprivation. (**A**) Five hours of sleep deprivation leads to a reduction in cofilin phosphorylation at serine 3 in the hippocampus. A representative blot is shown. Each band represents an individual animal. (n = 13–14, Student's t-test p=0.0090). (**B**) Mice were injected with pAAV9-CaMKIIα0.4-eGFP or pAAV9-CaMKIIα0.4-cofilinS3D-HA into the hippocampus to drive expression of eGFP or the mutant inactive form of cofilin (cofilinS3D) in excitatory neurons. This inactive mutant form of cofilin was made by substituting serine 3 for aspartic acid, which mimics a phosphoserine residue. An HA-tag was included to discriminate between mutant and

*Figure 3 continued on next page*

*Figure 3 continued*

endogenous cofilin. (**C**) A representative image showing that viral eGFP expression was restricted to the hippocampus. (**D–F**) Cofilin[S3D] expression was excluded from astrocytes in area CA1 as indicated by a lack of co-labeling (**F**) between viral cofilin (**D**) and GFAP expression (**E**). Scale bar, 100 μM. (**G**) Virally delivered cofilin[S3D] protein levels were approximately 75% (blue bar) of wild-type cofilin levels (green bar). Wild-type cofilin levels were not significantly affected by expression of cofilin[S3D]. An HA-tag antibody was used to detect the mutant inactive form of cofilin. (n = 4). (**H**) Hippocampal cofilin[S3D] expression prevents spine loss in apical/basal dendrites of CA1 neurons that is associated with sleep deprivation (n = 6, Student's t-test, p>0.05). (**I**) Hippocampal cofilin[S3D] expression prevents the decrease in apical/basal dendritic spine length in neurons of hippocampal that is caused by sleep deprivation (n = 6, Student's t-test, p>0.05). (**J**) Sleep deprivation does not alter the number of any spine type in apical/basal dendrites of CA1 neurons in the hippocampus of mice expressing cofilin[S3D] (n = 6, Student's t-test, p>0.05). (**K**) Sleep deprivation does not attenuate apical/basal spine density at any distance from the soma in mice expressing cofilin[S3D] (n = 6, Student's t-test, p>0.05). NSD: non-sleep deprived, SD: sleep deprived. Values represent the mean ± SEM. **p=0.0090. Student's t test. See also *Figure 3—figure supplement 1*. For separate analyses of apical and basal spine numbers see *Figure 3—figure supplement 2*.

The following source data and figure supplements are available for figure 3:

**Source data 1.** Sleep deprivation reduces cofilin phosphorylation in the hippocampus.

**Figure supplement 1.** Sleep deprivation does not alter cofilin phosphorylation in the prefrontal cortex.

**Figure supplement 1—source data 1.** Sleep deprivation does not alter cofilin phosphorylation in the prefrontal cortex.

**Figure supplement 2.** Cofilin[S3D] expression prevents sleep deprivation-induced reductions in spine numbers and dendrite length in both basal and apical dendrites of CA1 neurons.

Based on these findings, we hypothesized that suppressing cofilin activity would prevent the sleep deprivation-induced changes in spine numbers of CA1 neurons. To test this hypothesis, we used a phosphomimetic form of cofilin that renders it inactive, namely cofilin[S3D] (*Pontrello et al., 2012*; *Popow-Wozniak et al., 2012*; *Meberg et al., 1998*). Previous work suggested that cofilin[S3D] expression can inhibit endogenous cofilin activity (*Zhao et al., 2008*; *Shi et al., 2009*), through competition with endogenous cofilin for signalosomes where cofilin is activated by means of dephosphorylation (*Sarmiere and Bamburg, 2004*; *Konakahara et al., 2004*). For example, cofilin[S3D] may compete with endogenous cofilin for binding to the cofilin-dephosphorylating phosphatase slingshot (*Konakahara et al., 2004*). Importantly, cofilin[S3D] expression does not alter spine density under baseline conditions (*Pontrello et al., 2012*; *Shi et al., 2009*). We expressed either the phosphomimetic cofilin[S3D] or enhanced green fluorescent protein (eGFP), which served as a control, in hippocampal excitatory neurons of adult male C57BL/6J mice using Adeno-Associated Viruses (AAVs) (*Figure 3B,C*). A 0.4kb CaMKIIα promoter fragment was used to restrict expression to excitatory neurons (*Dittgen et al., 2004*). Virally mediated expression of cofilin[S3D] was observed in excitatory neurons in all 3 major sub-regions of the hippocampus three weeks after viral injection (*Figure 3D–F*). Western blot analyses of hippocampal lysates 3 weeks after injection showed that the level of virally delivered cofilin was roughly estimated 75% of the amount of endogenous wild-type cofilin and that the amount of wild-type cofilin per se was not substantially affected by expression of the mutant form (*Figure 3G*).

We subsequently determined whether expression of the inactive cofilin[S3D] prevented the loss of dendritic spines in hippocampal area CA1 caused by sleep deprivation. Analyses of Golgi-impregnated hippocampal neurons in area CA1 indicated that in cofilin[S3D] expressing mice sleep deprivation no longer reduced the spine density of apical/basal dendrites (NSD: 1.42 ± 0.03; SD: 1.34 ± 0.03; Student's t-test, p>0.05 *Figure 3H,J,K*; for separate apical and basal spine analyses see *Figure 3—figure supplement 2*) with the exception of a small but statistically significant reduction in branched spines of apical and basal dendrites (*Figure 3*, Figure Supplement C, D) and a decrease in number of spines on apical dendrites about 180 μm away from the soma (*Figure 3—figure supplement 2E,F*). Likewise, sleep deprivation no longer affected dendrite length (NSD: 1283.0 ± 35.95 μm, SD: 1250.1 ± 41.19 μm; Student's t-test, p=0.5612; *Figure 3I*, for separate apical and basal dendrite length analyses see *Figure 3—figure supplement 2B*). Together these data suggest that suppressing cofilin function in hippocampal neurons prevents the negative impact of sleep deprivation on spine loss and dendrite length of CA1 neurons.

# Suppressing cofilin function in hippocampal neurons prevents the impairments in memory and synaptic plasticity caused by brief periods of sleep deprivation

As a next step, we sought to determine whether prevention of the increase in cofilin activity in sleep-deprived mice would not only protect against the reduction in spine numbers on CA1 dendrites but also the functional impairment at the behavioral level. The consolidation of object-place memory requires the hippocampus (*Oliveira et al., 2010*; *Florian et al., 2011*) and is sensitive to sleep deprivation (*Havekes et al., 2014*; *Florian et al., 2011*; *Prince et al., 2014*). Therefore, we assessed whether cofilin[S3D] expression would prevent cognitive deficits caused by sleep deprivation in this task. Mice virally expressing eGFP or cofilin[S3D] were trained in this task 3 weeks after viral infection and sleep deprived for 5 hr immediately after training or left undisturbed in the home cage. Upon testing for memory the next day, sleep-deprived mice expressing eGFP showed no preference for the relocated object indicating that brief sleep deprivation impaired the consolidation of object-place memory. In contrast, mice expressing cofilin[S3D] showed a strong preference for the displaced object despite sleep deprivation (eGFP NSD: $45.2 \pm 6.4\%$, eGFP SD: $33.4 \pm 2.0\%$, cofilin[S3D] NSD: $51.9 \pm 2.9\%$, cofilin[S3D] SD: $53.2 \pm 4.6\%$; *Figure 4A*).

Expression of the mutant form of cofilin did not affect object exploration during training, exploration of an open field or zero maze indicating that anxiety levels were unaffected by expression of cofilin[S3D] in the hippocampus (*Figure 4—figure supplement 1A–C*). Moreover, using a behaviorally naïve cohort of mice, we found that cofilin[S3D] expression did not alter short-term object-place memory in the same task (*Figure 4—figure supplement 1D*). Together, these findings demonstrate that cofilin[S3D] expression specifically prevents the cognitive deficits caused by sleep deprivation. Although we can not rule out the possibility of off-target effects of the cofilin[S3D] mutant, we think that these are unlikely as expression of this mutant form of cofilin reversed the effects of sleep deprivation, restoring spine loss and memory to non-sleep deprived levels while not having an effect in non-sleep deprived mice.

To further define the role of cofilin in impairments in hippocampal function caused by sleep deprivation, we next determined if suppression of cofilin activity would prevent the deficits in hippocampal LTP caused by brief periods of sleep deprivation (*Havekes et al., 2012a*; *Abel et al., 2013*; *Vecsey et al., 2009*; *Prince et al., 2014*). Five hours of sleep deprivation significantly impaired long-lasting LTP induced by 4 high-frequency trains of electrical stimuli applied at 5-minute intervals (spaced 4-train stimulation) in hippocampal slices from mice expressing eGFP (*Figure 4B*), confirming our previously published findings with non-injected wild-type mice (*Vecsey et al., 2009*). In contrast, spaced 4-train LTP was unaffected by sleep deprivation in hippocampal slices from mice expressing the inactive cofilin[S3D] (*Figure 4C*). The expression of cofilin[S3D] or sleep deprivation did not alter basal synaptic properties or paired-pulse facilitation (*Figure 4—figure supplement 1E-H*) suggesting that the spine loss caused by sleep deprivation specifically impairs long-lasting forms of synaptic plasticity.

As a next step, we wanted to assess whether expression of a catalytically active version of cofilin (cofilin[S3A]) mimics the behavioral and synaptic plasticity phenotypes associated with sleep deprivation. Mice virally expressing eGFP or cofilin[S3A] were trained in the object-place memory task 3 weeks after viral infection and tested 24 hr after training. Mice expressing eGFP showed a strong preference for the relocated object while mice expressing cofilin[S3A] showed no preference for the object that was moved to a novel location (eGFP: $46.9 \pm 6.4\%$, cofilin[S3A]: $34.9 \pm 2.1\%$; *Figure 4—figure supplement 2B*). The observed memory deficit could not be explained by a reduction in object exploration during the training as the total object exploration time was similar for both groups during training (*Figure 4—figure supplement 2A*).

Based on these findings, we conducted a set of electrophysiological experiments to determine whether expression of cofilin[S3A] is also sufficient to induce impairments in spaced 4-train LTP. Cofilin[S3A] expression did not affect this form of L-LTP (*Figure 4—figure supplement 2E*). The expression of cofilin[S3A] did not alter basal synaptic properties or paired-pulse facilitation (*Figure 4—figure supplement 2C–D*).

In summary, these data show that phosphorylation-dependent reductions in cofilin activity in hippocampal excitatory neurons prevent the decrease in hippocampal spine numbers, and also prevent the functional impairments in synaptic plasticity and behavior caused by a brief period of sleep

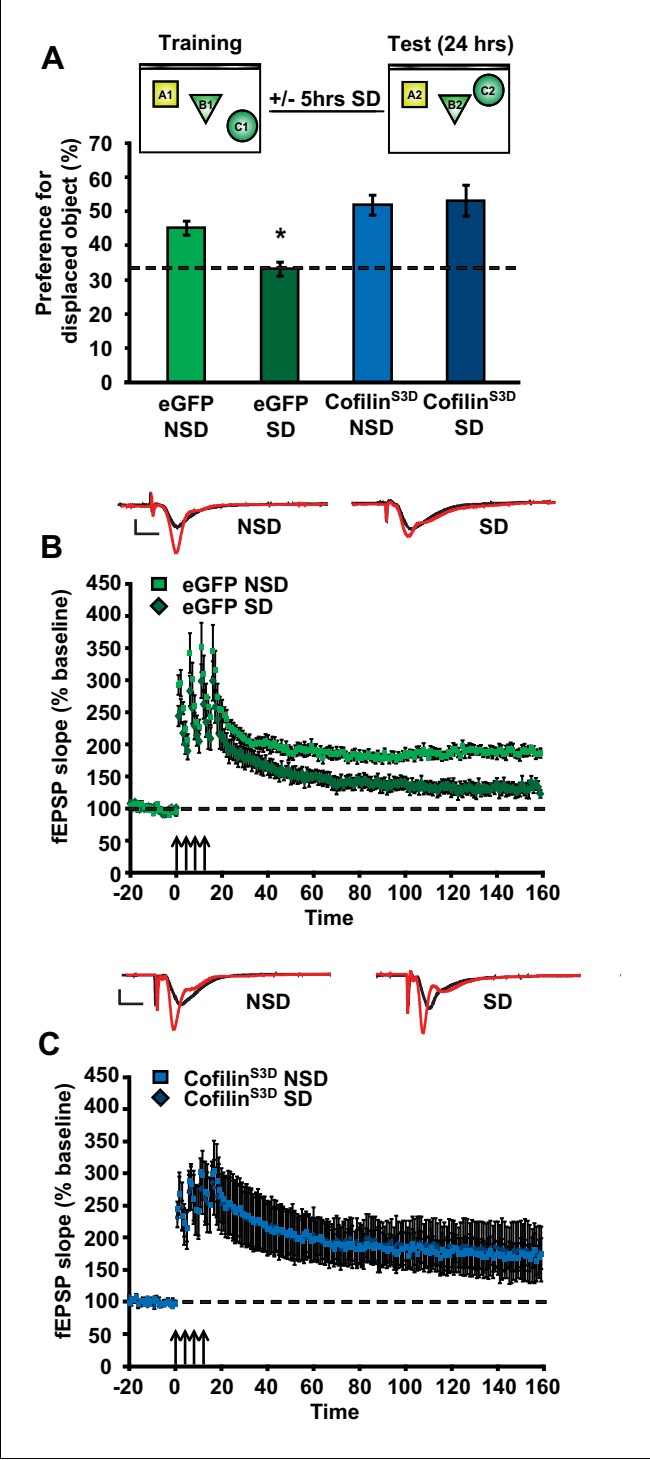

**Figure 4.** Increased cofilin activity in the hippocampus mediates the memory and synaptic plasticity deficits associated with sleep deprivation. (**A**) Mice expressing eGFP or cofilin[S3D] were trained in the hippocampus-dependent object-place recognition task. Half of the groups were sleep deprived for 5 hr and all mice were tested 24 hr later. Hippocampal cofilin[S3D] expression prevents memory deficits caused by sleep deprivation (n = 9–10, two-way ANOVA, effect of virus $F_{1,35}$ = 18.567, p=0.0001; effect of sleep deprivation $F_{1,35}$ = 2.975, p=0.093; interaction effect $F_{1,35}$ = 4.567, p=0.040; eGFP SD group versus other groups, p<0.05). The dotted line indicates chance performance (33.3%). (**B**, **C**) Following 5 hr of sleep deprivation, long-lasting LTP was induced in hippocampal slices by application of four 100 Hz trains, 1 s each, spaced 5 min apart to the Schaffer collateral pathway. Five hours of sleep deprivation impairs long-lasting LTP in slices from mice expressing eGFP (n = 6–7, *Figure 4 continued on next page*

*Figure 4 continued*

two-way ANOVA, effect of virus $F_{1,10}$ = 21.685, p<0.001). In contrast, virally delivered cofilin[S3D] prevents sleep deprivation-induced deficits (n = 5, two-way ANOVA, effect of virus $F_{1,8}$ = 0.016, p>0.902). NSD: non-sleep deprived, SD: sleep deprived. Values represent the mean ± SEM. *p<0.05 by posthoc Dunnet's test, **p<0.01 by Student's t test. See also *Figure 4—figure supplement 1*.

The following source data and figure supplements are available for figure 4:

**Source data 1.** Cofilin[S3D] expression prevents memory deficits in the object-location memory task caused by sleep deprivation.

**Figure supplement 1.** Cofilin[S3D] expression in hippocampal neurons does not affect exploratory activity, anxiety levels, or basal synaptic transmission.

**Figure supplement 1—source data 1.** Cofilin[S3D] expression in hippocampal neurons does not affect exploratory activity.

**Figure supplement 2.** Cofilin[S3A] expression in hippocampal neurons attenuates the formation of long-term object-location memories but not long-term potentiation induced by spaced-four train LTP.

deprivation. Furthermore, expression of constitutively active cofilin in hippocampal neurons is sufficient to mimic the memory deficits but not the synaptic plasticity impairment associated with a brief period of sleep deprivation.

## cAMP phosphodiesterase-4A5 (PDE4A5) causes the increase in cofilin activity associated with sleep deprivation through inhibition of the cAMP-PKA-LIMK pathway

Sleep deprivation attenuates cAMP signaling in the hippocampus through increased levels and cAMP hydrolyzing activity of PDE4A5 (*Vecsey et al., 2009*). Cofilin activity is known to be suppressed by the PKA-LIMK signaling pathway through the LIMK-mediated phosphorylation of cofilin at Ser-3 (*Lamprecht R, 2004*; *Nadella et al., 2009*). We hypothesized that the elevation in PDE4A5 activity, associated with sleep loss, could negatively impact the cAMP-PKA-LIMK signaling pathway by enhancing cAMP degradation, thereby leading to increased cofilin activity. Based on this hypothesis, we also anticipated that blocking PDE4A5 function in hippocampal neurons would make the cAMP-PKA-LIMK pathway, which controls cofilin activity, resistant to the effects of sleep deprivation. To test this hypothesis, we engineered a catalytically inactive form of PDE4A5 (referred to as PDE4A5[catnull]) in which an aspartate group located deep within the cAMP binding pocket of PDE4A5 (PDE4A5[D577A]), that is critical for catalytic activity, is replaced with an alanine group (*Baillie et al., 2003*; *McCahill et al., 2005*). Expression of PDE4A5[catnull] outcompetes the low levels of active, endogenous PDE4A5 from PDE4A5-containing signalosome complexes that specifically sequester it (*Houslay, 2010*), thereby preventing the breakdown of cAMP in the vicinity of those complexes. We used the viral approach (*Havekes et al., 2014*) to express PDE4A5[catnull] selectively in hippocampal neurons (*Figure 5A,B*). Four weeks after viral injections, expression of PDE4A5[catnull] was observed in all major hippocampal subregions (*Figure 5C–E*), and expression was excluded from astrocytes (*Figure 5F–H*). Expression of PDE4A5[catnull] did not alter PDE4 activity in the hippocampus, prefrontal cortex or cerebellum (*Figure 5—figure supplement 1A–C*). Next, we sleep deprived mice for 5 hr and assessed whether the phosphorylation of LIMK and cofilin was altered in the hippocampus. In agreement with our hypothesis, we observed that 5 hr of sleep deprivation reduced both LIMK and cofilin phosphorylation in hippocampal lysates from eGFP mice (*Figure 5I, J*). PDE4A5[catnull] expression prevented the sleep deprivation-induced decreases in LIMK and cofilin phosphorylation (*Figure 5I,J*). While expression of PDE4A5[catnull] fully restored the pcofilin/cofilin ratio in the hippocampus of sleep deprived mice to the levels observed under non-sleep deprivation conditions, it should be noted that phosphatases such as slingshot (*Sarmiere and Bamburg, 2004*) may also contribute to the reduction in cofilin phosphorylation levels under conditions of sleep deprivation. Three hours of recovery sleep was sufficient to restore both LIMK and cofilin

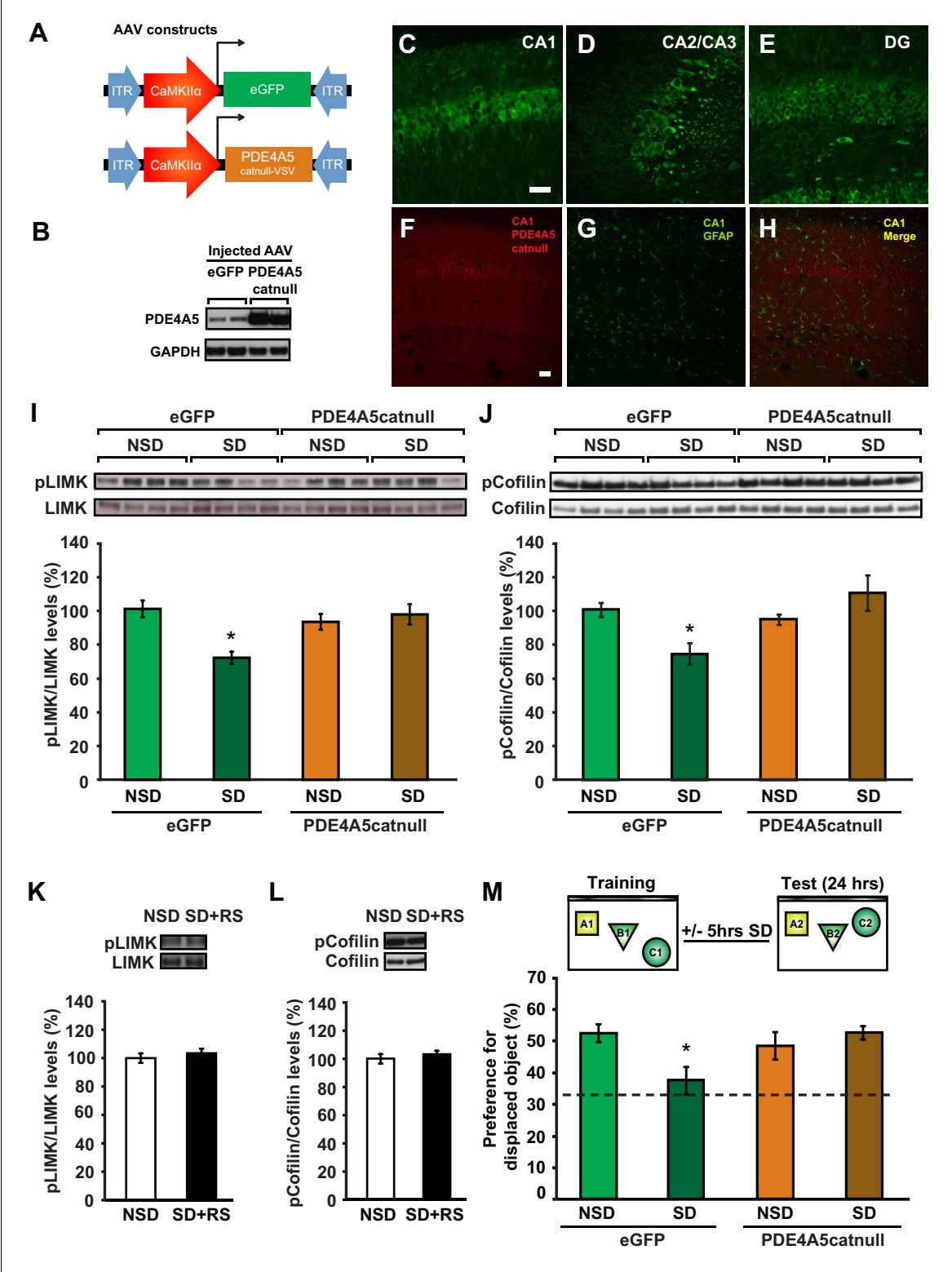

**Figure 5.** Expression of catalytically inactive PDE4A5 in hippocampal neurons prevents memory deficits and alterations in the cAMP-PKA-LIMK-cofilin signaling pathway associated with sleep deprivation. (A) Mice were injected with pAAV$_9$-CaMKIIα0.4-eGFP or pAAV$_9$-CaMKIIα0.4-PDE4A5$^{catnull}$-VSV into the hippocampus to drive neuronal expression of eGFP or catalytically inactive full-length PDE4A5 (PDE4A5$^{catnull}$). (B) Robust PDE4A5$^{catnull}$ expression was observed at the expected molecular weight, 108 kDa, in hippocampal lysates. (C–E) PDE4A5$^{catnull}$ expression was observed in all 3 subregions of the hippocampus. (F–H) PDE4A5$^{catnull}$ was not expressed in astrocytes reflected by a lack of co-labeling between PDE4A5$^{catnull}$ and GFAP expression. (I)
*Figure 5 continued on next page*

*Figure 5 continued*

Sleep deprivation causes a reduction in LIMK serine 596 phosphorylation in the hippocampus that is prevented by PDE4A5$^{catnull}$ expression (n = 7–8; two-way ANOVA, effect of virus $F_{1,27}$ = 3.299, p=0.08; effect of sleep deprivation $F_{1,27}$ = 6.124, p=0.02; interaction effect $F_{1,27}$ = 11.336, p=0.002; eGFP SD group versus other groups p<0.05). (J) Sleep deprivation causes a reduction in cofilin phosphorylation in the hippocampus that is prevented by PDE4A5$^{catnull}$ expression (n = 9–10; two-way ANOVA, effect of virus $F_{1,35}$ = 4.122, p=0.05; effect of sleep deprivation $F_{1,35}$ = 2.885, p=0.1; interaction effect $F_{1,35}$ = 9.416, p=0.004; eGFP SD group versus other groups p<0.05). (K, L) Three hours of recovery sleep after five hours of sleep deprivation restores hippocampal LIMK phosphorylation at serine 596 and cofilin phosphorylation at serine 3 to those observed in non-sleep deprived controls (p>0.45 for both comparisons). (M) Mice expressing eGFP or PDE4A5$^{catnull}$ were trained in the hippocampus-dependent object-place recognition task and immediately sleep deprived for 5 hr after training (SD) or left undisturbed (NSD). Hippocampal PDE4A5$^{catnull}$ expression prevents memory deficits caused by sleep deprivation (n = 8–10; two-way ANOVA, effect of virus $F_{1,33}$ = 2.626, p=0.115; effect of sleep deprivation $F_{1,33}$ = 2.311, p=0.138; interaction effect $F_{1,33}$ = 7.485, p=0.01; posthoc Dunnet test eGFP SD group versus other groups p<0.05). In all blots, each lane represents one individual animal. NSD: non-sleep deprived, SD: sleep deprived, SD+RS: sleep deprived plus recovery sleep. Scale bar, 100 μm. Values represent the mean ± SEM. *p<0.05 by posthoc Dunnet's posthoc test. See also *Figure 5—figure supplement 1*.

The following source data and figure supplements are available for figure 5:

**Source data 1.** Recovery sleep following sleep deprivation restores LIMK and cofilin phosphorylation levels in the hippocampus, and expression of an inactive version of PDE4A5 in hippocampal neurons prevents memory deficits associated with sleep deprivation.

**Figure supplement 1.** Expression of catalytically null PDE4A5 in the hippocampus: Catalytically inactive PDE4A5 without the unique N-terminal localization domain fails to prevent memory deficits associated with sleep loss.

**Figure supplement 1—source data 1.** Exploratory activity in mice expressing catalytically inactive PDE4A5 or PDE4A5Δ4 in hippocampal excitatory neurons.

phosphorylation levels in the hippocampus (*Figure 5K,L*). The latter observation is in line with our previous observations that a few hours of recovery sleep is sufficient to restore hippocampal synaptic plasticity (*Vecsey et al., 2009*).

## Blocking PDE4A5 function in hippocampal neurons prevents memory deficits caused by sleep deprivation

Because PDE4A5$^{catnull}$ expression in hippocampal neurons prevents changes in the cAMP-PKA-LIMK-cofilin pathway caused by sleep deprivation, we hypothesized that expression of PDE4A5$^{catnull}$ in hippocampal excitatory neurons would also prevent the memory deficits induced by 5 hr of sleep deprivation. Mice expressing eGFP showed a clear preference for the displaced object 24 hr after training, which was lost in animals that were deprived of sleep for 5 hr immediately after training (*Figure 5M*). In contrast, mice expressing PDE4A5$^{catnull}$ showed a strong preference for the displaced object despite sleep deprivation (*Figure 5M*). The memory rescue was not a result of altered exploratory behavior during training in the object-place recognition task (*Figure 5—figure supplement 1D*). Furthermore, PDE4A5$^{catnull}$ expression did not alter anxiety levels and exploratory behavior in the open field (*Figure 5—figure supplement 1E*).

Although the catalytic unit of the 25 distinct PDE4 isoforms is highly conserved, each has a unique N-terminal localization sequence that directs isoform targeting to a specific and unique set of protein complexes (signalosomes) (*Houslay, 2010*). This allows for a highly orchestrated sequestering of cAMP signaling in specific intracellular domains rather than a general, global degradation of cAMP throughout the cell (*Houslay, 2010*). We therefore aimed to determine whether the rescue of memory impairments by expression of PDE4A5$^{catnull}$ requires the unique N-terminal domain of PDE4A5. To answer this question, we engineered a truncated version of PDE4A5$^{catnull}$ that lacks the first 303 base pairs encoding the isoform unique N-terminal domain (*Bolger et al., 2003*) (referred to as PDE4A5$^{catnullΔ4}$, *Figure 5—figure supplement 1F*) and expressed this mutant in excitatory neurons in the hippocampus using a viral approach. As this species has no targeting N-terminus then, unlike the full length inactive PDE4A5 that displaces endogenous active PDE4A5 from its functionally relevant location in the cell and thereby increase cAMP levels localized to the sequestering signaling complex, this engineered 5' truncated complex would simply lead to the expression of an inactive PDE4A catalytic unit unable to be targeted like the native enzyme and so unable to exert an effect on localized cAMP degradation in the functionally relevant compartment.

Western blot analyses of hippocampal tissue 4 weeks after viral injection confirmed the presence of the truncated PDE4A5$^{catnull\Delta4}$ at the protein level using an antibody that detects all PDE4A isoforms and an antibody against the HA-tag (*Figure 5—figure supplement 1F,G*). With a behaviorally naïve cohort of mice now expressing eGFP or PDE4A5$^{catnull\Delta4}$ we repeated the object-place recognition task. Brief sleep deprivation after training in the object-place recognition task resulted in a loss of preference for the displaced object in mice expressing PDE4A5$^{catnull\Delta4}$ (*Figure 5—figure supplement 1I*). The inability of PDE4A5$^{catnull\Delta4}$ to prevent the memory deficit caused by sleep deprivation was not a consequence of altered exploration levels during training (*Figure 5—figure supplement 1H*). This finding indicates that the memory rescue in the previous experiment was a result of the full length PDE4A5$^{catnull}$ being sequestered to specific signalosomes through the isoform-unique N-terminal region rather than a consequence of PDE4A5$^{catnull}$ being unable to target the functionally relevant complexes sequestering full length PDE4A5. It also indicates that displacing sequestered, active endogenous PDE4A5 in hippocampal excitatory neurons is sufficient to prevent memory deficits induced by 5 hr of sleep deprivation. Overall, these data suggest that sleep deprivation negatively impacts spine numbers by targeting the PKA-LIMK-cofilin pathway through the alterations in activity of PDE4A5 (*Figure 6*).

## Discussion

One of the major challenges in sleep research is the elucidation of molecular mechanisms and cellular circuits underlying the adverse consequences of sleep loss. Here, we use in vivo rescue experiments to define a critical molecular mechanism by which brief sleep deprivation leads to cognitive

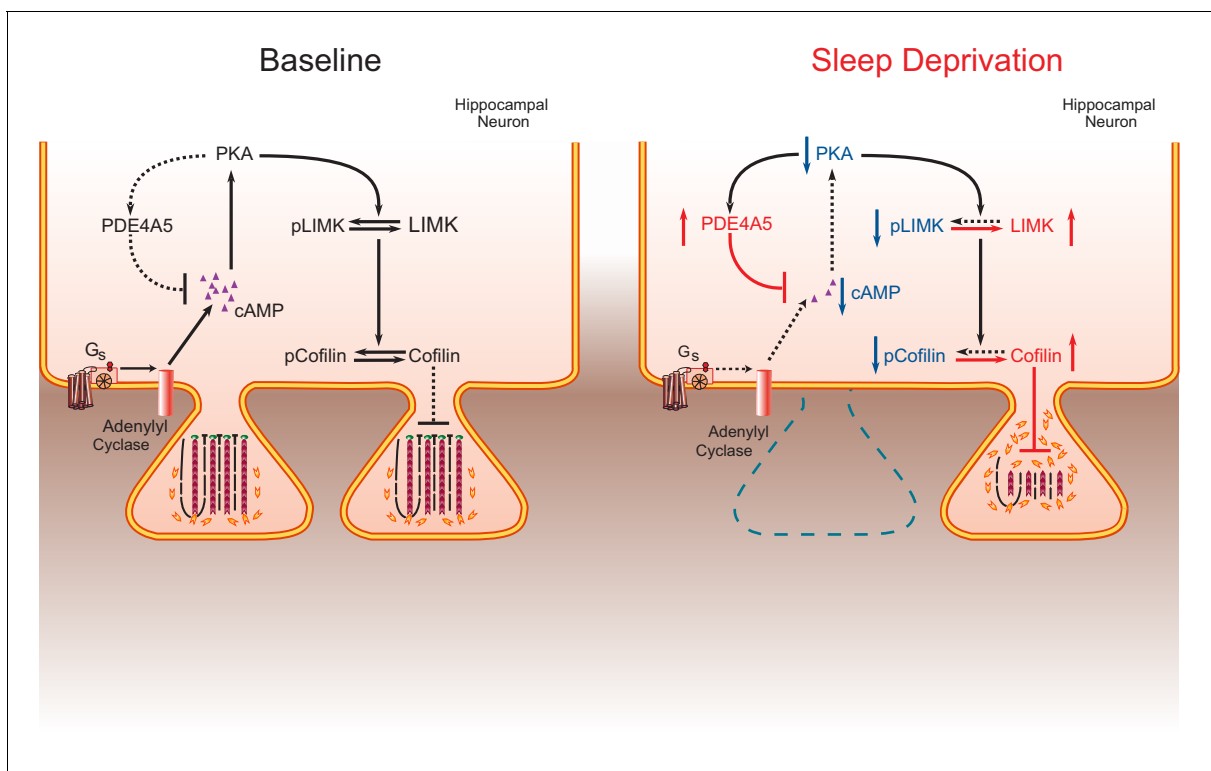

**Figure 6.** The impact of sleep deprivation on hippocampal spine dynamics. Sleep deprivation increases PDE4A5 protein levels that cause a reduction in cAMP levels and attenuation of the PKA-LIMK signaling pathway, which results in a reduction in the phosphorylation of cofilin. Dephosphorylated cofilin can lead to spine loss. Suppressing PDE4A5 function through viral expression of a catalytically inactive PDE4A5 prevents alterations in LIMK and cofilin signaling as well as the cognitive impairments caused by sleep deprivation. Likewise, attenuating cofilin activity through viral expression of a catalytically inactive form of cofilin prevents the loss of dendritic spines, impairments in synaptic plasticity, and memory deficits associated with sleep loss. Proteins whose function is reduced after sleep deprivation are shown in blue. Proteins whose function is promoted by sleep deprivation are shown in red.

impairments. First, we show that sleep deprivation dramatically reduces spine number and dendrite length of hippocampal CA1 neurons without affecting dendritic structure of CA3 neurons. Second, we demonstrate that sleep deprivation increases cofilin activity in the hippocampus, but not the prefrontal cortex, which is a likely explanation for the reductions in CA1 spine numbers and dendrite length. Third, we find that three hours of recovery sleep restores spine number, dendrite length and cofilin phosphorylation levels to those observed in non-sleep deprived mice. Fourth, we show that suppression of cofilin activity in hippocampal excitatory neurons is sufficient to prevent sleep-deprivation-induced decreases in dendritic spines number, LTP impairments, and memory. Fifth, we demonstrate that hippocampal expression of constitutively active cofilin is sufficient to cause long-term memory deficits but not LTP impairments. Sixth, we find that suppression of PDE4A5 function through overexpression of a catalytically inactive mutant version of PDE4A5 changes LIMK and cofilin phosphorylation levels caused by sleep deprivation. Finally, we show that hippocampal suppression of PDE4A5 function prevents the negative impact of sleep deprivation on memory consolidation. Thus, our studies demonstrate that changes in the cAMP/PKA/LIMK/cofilin pathway are necessary to cause memory deficits under conditions of sleep deprivation.

In light of the fact that elevated cofilin activity can lead to spine shrinkage and spine loss (*Zhou et al., 2004*; *Pontrello et al., 2012*), our genetic manipulations of cofilin and PDE4A5 signaling independently link impairments in synaptic plasticity and memory caused by brief sleep deprivation with the loss of dendritic spines in the hippocampus. Deficits in synaptic plasticity and memory both represent read outs of the impact of sleep deprivation on hippocampal function, but our work does not directly examine the direct relationship between synaptic plasticity and memory, a topic that has been the subject of extensive study and discussion in the literature (*Lynch, 2004*; *Sah et al., 2008*). That care should be taken to directly relate LTP deficits with memory impairments is emphasized by our findings that expression of constitutively active cofilin is sufficient to cause memory deficits while it does not impact at least one form of L-LTP that is disrupted by sleep deprivation (*Vecsey et al., 2009*).

We observed a significant reduction in the total number of dendritic spines of excitatory CA1 neurons after 5 hr of sleep deprivation. This substantial decrease in dendritic spine number occurs rapidly and exceeds the fluctuations in hippocampal spine number observed across the estrus cycle (*González-Burgos et al., 2005*), or the changes in spine number caused by stress (*McLaughlin et al., 2005*; *Shors et al., 2001*). In contrast to sleep deprivation, acute stress results in an increase rather than a decrease in CA1 spine number in male rats (*Shors et al., 2001*). Even 3 to 4 weeks of chronic stress or systemic delivery of corticosterone does not alter dendritic arborization of CA1 neurons (*McLaughlin et al., 2005*). It is also unlikely that other factors associated with the procedure to keep animals awake rather than sleep deprivation per se causes the spine loss as our previous work indicated that applying the exact same amount of stimulation in the waking phase (*i. e.* the dark phase) does not lead to memory impairments (*Hagewoud et al., 2010a*).

In line with our finding of reductions in spines during sleep deprivation, work by Yang and colleagues revealed that sleep promotes dendritic spine formation in neurons activated by learning (*Yang et al., 2014*). Combined with our work, these experiments suggest that sleep deprivation disrupts learning-induced changes in spines that occur during sleep. Importantly, our structural studies reveal that spine loss is reversed by recovery sleep, consistent with this idea. Thus, our work reveals a distinct, selective, and rapid effect of brief periods of sleep loss on synaptic structure. It is noteworthy that even a short period of sleep deprivation acts to trigger such a dramatic effect on neuronal structure, which is reversed by recovery sleep.

Studies assessing the impact of sleep deprivation on electrophysiological properties of excitatory hippocampal neurons suggest that sleep deprivation negatively impacts long-lasting forms of LTP (*Havekes et al., 2012a*; *Abel et al., 2013*). In this study and our previous work (*Vecsey et al., 2009*; *Prince et al., 2014*), we showed that 5 hr of sleep deprivation attenuates long-lasting forms of LTP in the hippocampus. We observed that expression of an inactive mutant form of cofilin prevented the reductions in CA1 spine number, the impairment in a long-lasting form of LTP caused by sleep loss. It is interesting to note that three hours of recovery sleep not only restores spine numbers in CA1 neurons, but also hippocampal LIMK and cofilin phosphorylation levels. These findings complement our previous electrophysiological studies, in which we showed that such a short period of recovery sleep also restores deficits in LTP caused by 5 hr of sleep deprivation (*Vecsey et al., 2009*).

Our work reveals that PDE4A5 is a critical mediator of the impact of sleep deprivation on memory consolidation. Indeed, one reason why hippocampal area CA1 is specifically vulnerable to sleep deprivation may be the high level of PDE4A5 expression in this region (*McPhee et al., 2001*). Specific PDE4 isoforms are sequestered by distinct signalosome complexes that regulate localized cAMP signaling and impart functionally distinct roles (*Houslay, 2010*). Impairing the function of PDE4A5 signalosomes through expression of a full length catalytically inactive form of PDE4A5 exerts a dominant negative action, phenotypically identified here as preventing the alterations in LIMK and cofilin signaling caused by sleep deprivation. This makes memory consolidation resistant to the negative impact of sleep loss. Consistent with the notion that a key functional role of the isoform-unique N-terminal region of PDE4 isoforms is the targeting to signalosomes so as to exert functionally distinct actions (*Houslay, 2010*), the hippocampal expression of a catalytically in active version of PDE4A5 lacking the isoform unique N-terminal domain fails to rescue the cognitive deficits associated with sleep loss. The latter observation suggests that the isoform-specific N-terminal domain of PDE4A5 targets this specific PDE isoform to signalosomes that degrade cAMP in the vicinity of complexes that are particularly sensitive to sleep deprivation such as the complexes that contain LIMK and cofilin. Consistent with this, no such dominant negative phenotype is evident in a catalytically inactive PDE4A construct engineered to lack such an N-terminal targeting region.

Our data contradict the synaptic homeostasis hypothesis for sleep function. This hypothesis proposes that sleep functions to downscale synaptic strength that has increased as a result of neuronal activity and experiences during wakefulness (*Tononi and Cirelli, 2006*). This hypothesis has focused on explaining data from the cortex rather than the hippocampus, but one previously published study has suggested that the synaptic homeostasis hypothesis applies to the hippocampus as well (*Vyazovskiy et al., 2008*). However, the hippocampus may be unique from the cortex as the hippocampus is involved in episodic memory and in much greater experience-dependent plasticity than anywhere else in the brain and thus our findings may not extend to other areas where synaptic plasticity is not as prominent. Further, the hippocampus also exhibits many distinct forms of synaptic plasticity. Here, we examine structural changes in hippocampal neurons and find that extended wakefulness leads to a loss of synaptic spines mediated by a signaling pathway involving cofilin. This suggests that prolonged wakefulness down-regulates synaptic connectivity in the hippocampus. As little as 3 hr of recovery sleep is sufficient to restore signaling through these complexes, suggesting that sleep functions to restore synaptic connectivity. Thus, the signaling pathways that mediate changes in dendritic structure are rapidly impaired by sleep loss and then can be quickly restored during recovery sleep.

Lack of sleep is a common problem in our 24/7 modern society and it has severe consequences for health, overall wellbeing, and brain function (*Bryant et al., 2004*; *Harrison and Horne, 2000*). Despite decades of research, the mechanisms by which sleep loss negatively impacts brain function have remained unknown. Our findings suggest that the cognitive impairments caused by brief sleep deprivation are a result of altered spine dynamics leading to a reduction in spine numbers. Our findings may also explain the reduction in hippocampal volume observed in an animal model of more chronic sleep restriction (*Novati et al., 2011*) and sleep disorders, such as primary insomnia (*Riemann et al., 2007*) as well as sleep apnea (*Morrell et al., 2003*). Our work defining the molecular pathway through which sleep deprivation impacts memory consolidation underscores the importance of the plasticity of the neuronal cytoskeleton and reveals that rapid synaptic remodeling occurs with changes in behavioral state.

## Materials and methods

### Subjects

Experimentally naïve C57BL/6J male mice (2–3 months of age; IMSR_JAX:000664) were obtained from Jackson laboratories at an age of 6 weeks and housed in groups of 4 with littermates on a 12 hr/12 hr light/dark schedule with lights on at 7 am (ZT0). Mice had food and water available *ad libitum*. In case of the viral studies, mice underwent surgery at an age of 8–12 weeks, were single housed for 5 days and then pair-housed with a littermate throughout the experiment. For the perfusion experiments, mice were single housed 1 week prior to the start of the experiment. For all

experiments, mice were randomly assigned to groups and were handled for 5 days for 2 min per day.

## Sleep deprivation

Mice were sleep deprived using the gentle stimulation method (*Vecsey et al., 2009*; *Hagewoud et al., 2010a*; *Vecsey et al., 2013*; *Hagewoud et al., 2010b*, *2010c*). In short, animals were kept awake by gentle tapping the cage, gently shaking the care and/or removing the wire cage top. Their bedding was disturbed in cases when mice did not respond to tapping or shaking the cage. This method of sleep deprivation has been validated by our laboratory using EEG recordings (*Meerlo et al., 2001*).

## Behavioral assays

In the object-place recognition task, mice learn the location of 3 distinct objects and were tested for memory of the object locations 24 hr after training by displacing one of the objects. Training commenced at ZT0 or 45 min after lights on using the previously described training protocol (*Oliveira et al., 2010*; *Tretter et al., 2009*). Mice were trained 3 or 4 weeks after viral surgery. Object exploration levels were scored manually by the experimenter blind to treatment conditions. The zero maze and open field studies were conducted as previously described (*Tretter et al., 2009*; *Havekes et al., 2012b*).

## Viral surgeries

Mice were anaesthetized using isoflurane and remained on a heating pad throughout the surgery and kept warm using a heating lamp for 5–10 min during the recovery from the anesthesia until the mouse was awake. Mice received metacam and buprenol as analgesics during and post-surgery and artificial tears (Puralube) were used to prevent the eyes from drying out during surgery. Two small holes were drilled in the skull at the appropriate locations using a microdrill. The virus was injected using a nanofil 33G beveled needles (WPI) attached to a 10 µl Hamilton syringe. A microsyringe pump (UMP3; WPI) connected to a mouse stereotax and controller (Micro4; WPI) were used to control the speed of the injections. The needle was slowly lowered to the target site over the course of 3 min and remained at the target site for 1 min before beginning of the injection (0.2 µl per minute). After the injection, the needle remained at the target site for 1 min and then was slowly gradually removed over a 5 min period. The coordinates for the bilateral injections are (A/P −1.9 mm, L/M ± 1.5 mm, and 1.5 mm below bregma). After removal of the needle, a small amount of bone wax (Lukens) was used to close the drill holes and the incision was closed with sutures.

## DNA manipulation and Virus constructs

Site-directed mutagenesis of plasmid DNA was carried out to generate PDE4A5$^{catnull}$ using the Stratagene QuikChange Site-Directed Mutagenesis kit, using the method in the manufacturer's instructions. N-terminal lacking PDE4A5$^{catnull}$ was generated using standard PCR cloning procedures and the Stratagene PfuUltra High-Fidelity DNA polymerase. Purified plasmid DNA was produced using Qiagen QIAprep kits and stored at 4°C. The pAAV$_9$-CaMKIIα0.4-PDE4A5$^{catnull}$-VSV, pAAV$_9$-CaMKIIα0.4-PDE4A5$^{catnullΔ4}$-HA, pAAV$_9$-CaMKIIα0.4-Cofilin$^{S3D}$-HA, pAAV$_9$-CaMKIIα0.4-Cofilin$^{S3A}$-HA and pAAV$_9$-CaMKIIα0.4-eGFP were constructed by standard methods and packaged by the University of Pennsylvania viral core. Transduced cofilin$^{S3D}$ may compete with endogenous cofilin for binding to the cofilin-specific phosphatase slingshot, thereby leading to inactivation of the endogenous protein (*Sarmiere and Bamburg, 2004*; *Konakahara et al., 2004*). Titers ranged from 2.4 × $10^{12}$ to 4.91 × $10^{13}$ genome copy numbers. A 0.4kb CaMKIIα promoter fragment (*Dittgen et al., 2004*) was used to restrict expression to excitatory neurons. An HA-tag was included to discriminate endogenous from virally expressed proteins. Approximately 1 µl, (corrected for genome copy number between constructs) was injected per hippocampus.

## Biochemistry

The cAMP-specific PDE activity assays and western blots to assess sleep deprivation-induced changes in PDE4A5 levels were conducted as described (*Vecsey et al., 2009*). Hippocampal tissue was lysed using a tissue ruptor (Qiagen, Germany) in lysis buffer (Tris 50 mM, pH: 9, sodium

deoxycholate 1%, Sodium fluoride 50 mM, activated sodium vanadate 20 μM, EDTA 20 μM, and beta-glycerophosphate 40 μM. Additional phosphatase inhibitor cocktail (Thermo scientific) and protease inhibitors (Roche, Switzerland) were added to the freshly prepared buffer just prior to tissue lysis. Samples were centrifuged for 10 min at 13,000 ×g at 4°C and supernatant was collected. Protein concentration of the samples was measured using the Bradford method (Biorad, Hercules ,CA, USA) and sample concentration was corrected using additional lysis buffer. Afterwards LDS sample buffer (Nupage, Invitrogen, Carlsbad, CA, USA) including 2-mercaptoethanol was added and samples were for boiled for 5 min prior to loading on Criterion TGX 18-well 4–20% gels. After electrophoreses, proteins were transferred to PVDF membrane followed by blocking for 1 hr in 5% milk in TBST or 5% BSA in TBST (in case of cofilin antibodies). After blocking, the following antibodies were used GAPDH (1:1000, Santa Cruz, Santa Cruz, CA, USA RRID:AB_10167668), PDE4A (1:1000(27)), PDE4A5 (1:1000(27)), pCofilin (1:1000, Cell signaling, RRID:AB_2080597), Cofilin (1:3000 BD Transduction Laboratories,San Jose, CA, USA; RRID:AB_399515), HA-tag (1:1000, Roche, RRID:AB_ 390918), VSV-G tag (1:1000, Abcam, United Kingdom, RRID:AB_302646), LIMK (1:2000, Millipore, Billerica, MA, USA; RRID:AB_1977324). Polyclonal phospho-serine 596 LIMK antibody was generated by New England Biopeptides (Gardner, MA, USA) using CDPEKRP(pS)FVKLEQ peptide. After incubation with the primary antibodies, membranes were incubated in HRP-conjugated secondary antibodies for 1 hr at room temperature (Santa Cruz, mouse secondary antibody 1:1000, RRID:AB_ 641170; Santa Cruz, rabbit secondary antibody, RRID:AB_631746). The immunoreactive bands were captured on autoradiography film (Kodak, Rochester, NY, USA) and analyzed using ImageJ (NIH).

## Immunohistochemistry

Immunohistochemistry was conducted as described previously (*Havekes et al., 2012b*; *Isiegas et al., 2008*). In short, animals were transcardially perfused with ice cold 4% paraformaldehyde in PBS followed by a 48 hr post fixation in 4% PFA. Coronal brain sections were cut at a thickness of 25 microns. Sections were rinsed in PBS, blocked with 5% normal serum and incubated in PBS with 0.1% triton and 2% normal serum with either of the following antibodies or combinations of antibodies PDE4A5 (1:200, (27)), HA-tag (1:200, Roche, RRID:AB_390918), VSV-G tag (1:2000, Abcam, RRID:AB_302646), GFAP-alexa 488 (1:200, Invitrogen, RRID:AB_143165) followed by the appropriate Alexa fluor-conjugated secondary antibodies (1:1000 Invitrogen, RRID:AB_141459, RRID:AB_10562718, RRID:AB_10564074). Fluorescent images were analyzed using a Leica confocal microscope.

## Diolistic staining

After sleep deprivation mice were injected (i.p) with a lethal dose of morbital and perfused with phosphate-buffered saline (PBS, 3 min at RT), followed by 1.5% PFA (20 min at RT). Brains were then removed and post-fixed in 1.5% PFA (40 min at RT). After post perfusion incubation in 1.5% PFA, each hemisphere was cut in 130 μm slices using a vibratome. Slices were collected to multiwell plates filled with PBS. After one hour incubation in room temperature, PBS was removed and slices were stained with a GeneGun (Biorad, pressure: 100–120 psi) using nylon filter (Merc Millipore, 10 μm, cat. No. NY1004700). DiI bullets were prepared as described (*Seabold et al., 2010*). After staining, slices were incubated over night at RT in PBS. The next day slices were incubated for one hour in 4% PFA and mounted with DapiFluoromount G (SouthernBiotech). Microphotographs of DiI stained apical dendrites in the stratum radiatum of CA1 area (approx. 100 μm from cell bodies) were performed in z-stacks using Zeiss LSM 780 (step 0.3 mm, objective 63x, digital magnifications 5x, resolution 1024 × 1024). Linear density (per mm of dendrite) and size of spines were counted using SpineMagick software. On average 155 spines were analyzed per an animal. Importantly, for the comparison of the spine numbers using golgi and diolistic staining methods we focused *specifically* on the second and third branch of the apical dendrites as the diolistic staining technique is suboptimal to label and analyze branches farther away from the soma.

## Electrophysiology

Experiments were performed in the hippocampal Schaffer collateral pathway as previously described (*Vecsey et al., 2009*; *Havekes et al., 2012b*). Briefly, male mice injected with eGFP or cofilin[S3D] virus were sacrificed by cervical dislocation, and hippocampi were quickly collected in chilled,

oxygenated aCSF containing 124 mM NaCl, 4.4 mM KCl, 1.3 mM $MgSO_4 \times 7H_2O$, 1 mM $NaH_2PO_4 \times H_2O$, 26.2 mM $NaHCO_3$, 2.5 mM $CaCl_2 \times 2H_2O$, and 10 mM D-glucose bubbled with 95% $O_2$ / 5% $CO_2$. 400 µm thick transverse hippocampal slices were placed in an interface recording chamber at 28°C (Fine Science Tools, Foster City, CA). Slices were equilibrated for at least 2 hr in aCSF (pH 7.4). The stimulus strength was set to elicit 40% of the maximum field excitatory postsynaptic potential (fEPSP) amplitude. The average of the baseline initial fEPSP slope values over the first 20 min was used to normalize each initial fEPSP slope.

## Golgi analyses

Brains were impregnated using the Rapid Golgi stain kit (FD Neurotechnologies Inc) according to the instructions. Coronal sections (80-*um* thickness) that covered the rostro-caudal axis of CA1 of the hippocampus were analyzed. The serial sections were then chosen and analyzed using a stereology-based software (Neurolucida, v10, Microbrightfield, VT), and Zeiss Axioplan 2 image microscope with Optronics MicroFire CCD camera (1600 × 1200) digital camera, motorized X, Y, and Z-focus for high-resolution image acquisition and digital quantitation in combination with a 100x objective using a sophisticated and well established method that should represent a 3D quantitative profile of the neurons sampled and prevents a failure to detect less prominent spines.

Our sampling strategy is to prescreen the impregnated neurons along the anterior/posterior axis of the region of interest to see if they were qualified for analysis. Neurons with incomplete impregnation or neurons with truncations due to the plane of sectioning were not collected. Moreover, cells with dendrites labeled retrogradely by impregnation in the surrounding neuropil were excluded. We also made sure there was a minimal level of truncation at the most distal part of the dendrites; this often happens in most of the Golgi studies, likely due to the plane of sectioning at top and bottom parts of the section. The brains were cut at a 80 µm thickness. With consideration of the shrinkage factor after processing (generally 10–25% shrinkage), the thickness of the section is even less, so the visualization of the spine subclass is no issue as we used a 100x Zeiss objective lens with immersion oil, which is sufficient to resolve the details or subtype of the spines for laborious counting. All analyses were conducted by an experimenter blind to treatment.

## Statistics

Behavioral and electrophysiological data were analyzed using Student's t-tests or two-way ANOVAs (in some cases with repeated measures as the within subject variable). Dunnett's tests were used for post-hoc analyses where needed. Biochemical data was analyzed using independent samples t-tests. The experimenter was blind to group treatment in all studies. Differenceswere considered statistically significant when $p < 0.05$. All data are plotted as mean ± s.e.m.

## Acknowledgements

We thank Abel lab members for their help with these experiments and for their comments on the manuscript. We thank James Bamburg (Colorado State University) for providing cofilin antibodies. We thank Amita Sehgal (University of Pennsylvania), Tom Jongens (University of Pennsylvania) and Noreen O'Connor-Abel (University of Pennsylvania) for input on a previous version of the manuscript. We thank Paul Schiffmacher for assistance with the illustrations.

## Additional information

### Funding

| Funder | Grant reference number | Author |
|---|---|---|
| Nederlandse Organisatie voor Wetenschappelijk Onderzoek | postdoctoral fellowship 825.07.029 | Robbert Havekes |
| University of Pennsylvania | UPENN rsearch foundation grant | Robbert Havekes Ted Abel |
| National Institutes of Health | postdoctoral fellowship, 5K12GM081529 | Jennifer C Tudor |

| National Institutes of Health | postdoctoral fellowship, T32 NS077413 | Sarah L Ferri |
| --- | --- | --- |
| European Commission | FP7-PEOPLE-2009-RG-Alco_CaMK | Kasia Radwańska |
| National Science Centre | Harmonia 2013/08/m/NZ3/00861 Grant | Kasia Radwańska |
| Medical Research Council | Grant MR/J007412/1 | George S Baillie |
| National Institutes of Health | 1RO1MH086415 | Ted Abel |
| National Institutes of Health | RO1AG017628 | Ted Abel |

The funders had no role in study design, data collection and interpretation, or the decision to submit the work for publication.

## Author contributions

RH, Manuscript was prepared, Immunohistochemistry experiments were carried out, Biochemical and golgi experiments were carried out, Behavioral experiments were carried out, Experiments were conceived and designed, Analysis and interpretation of data; AJP, VGL, Electrophysiological experiments were carried out, Analysis and interpretation of data; JCT, SLF, JPD, Biochemical and golgi experiments were carried out, Analysis and interpretation of data; RTH, VMB, SGP, KR, Acquisition of data, Analysis and interpretation of data; SJA, PM, MDH, GSB, Conception and design, Drafting or revising the article; TA, Manuscript was prepared, Experiments were conceived and designed

## Author ORCIDs

Jennifer C Tudor, http://orcid.org/0000-0002-3826-3012
Ted Abel, http://orcid.org/0000-0003-2423-4592

## Ethics

Animal experimentation: This study was performed in strict accordance with the recommendations in the Guide for the Care and Use of Laboratory Animals of the National Institutes of Health. All of the animals were handled according to approved institutional animal care and use committee (IACUC protocols 804240, 804407, 802784) of the University of Pennsylvania and Head Necki Institute of Experimental Biology, Warsaw.

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
