## [Decision Letter]

Thank you for submitting your work entitled "Sleep deprivation causes memory deficits by negatively impacting neuronal connectivity in hippocampal area CA1" for consideration by *eLife*. Your article has been reviewed by two peer reviewers, and the evaluation has been overseen by Joseph Takahashi as the Reviewing Editor and a Senior Editor. The following individual involved in review of your submission has agreed to reveal their identity: Robert Greene (peer reviewer).

The reviewers have discussed the reviews with one another and the Reviewing Editor has drafted this decision to help you prepare a revised submission.

Summary:

In this study Havekes et al. investigated the cellular mechanisms underlying sleep deprivation induced learning deficits. They found that 5 hours of sleep deprivation can cause a large reduction in hippocampal CA1 spine density, which can be reversed with 3 hours of recovery sleep. Sleep deprivation also causes an increase in the activity of cofilin, which is known to regulate spine morphology. Suppression of cofilin activity prevents sleep deprivation induced spine loss, synaptic plasticity, and memory deficit. Reducing the activity of PDE4A5, an upstream regulator of cofilin activity has similar effects.

This study examines a series of cellular processes involved in structural synaptic plasticity and their role in sleep deprivation induced learning deficit, thus providing important insights into how sleep regulates learning and memory.

Essential revisions:

1) The causal association of SD effects on PDE and synaptic structure, LLTP and long-term memory are somewhat unclear. The experimental protocol involves training, immediately followed by 5h SD and a probe, 24h after that. This implies that LLTP disruption occurs at least an hour or more after induction of synaptic potentiation or that synaptic potentiation in CA1 normally occurs following the training, during sleep. Thus SD following training might disrupt the putative sleep potentiation as proposed. If the latter is occurring, then synaptic strength, dendritic structure and spine number should increase accordingly from the time immediately following training to 5h following training.

Alternatively, reduction of cAMP, PKA or cofilin inactivation should reduce LLTP an hour or more after LTP induction.

Another possibility is that the cAMP/PKA/cofilin SD associated changes and spine/dendritic structure, LLTP and/or memory changes are not mechanistically linked. Evidence for such linkage would seem essential to the proposed mechanisms of SD induced memory deficits.

The primary conclusion drawn from this study is that 5h of SD reduced CA1 spine density and LLTP of the Schaffer Collateral synapses in association with hippocampal dependent learning and memory dysfunction, by increasing cofilin activity. The evidence presented includes 1) western blots showing 5h SD decreases the ratio of the pCofilin-inactive/cofilin-active form and 2) over-expression of an inactive mutant cofilin^S3D^ in CA1 prevented SD effects on spines, LLTP and L&M.

These data are preliminary in the sense that there is no evidence that cofilin activity was affected (or to what degree) by the cofilin^S3D^, there is no direct measure of the putative SD effect on cofilin and it is not clear how great an effect on cofilin activity is needed (i.e. is sufficient) to elicit the observed SD effect on spines, LLTP and L&M. Can off-target effects of the mutant be ruled out? Does a phospho-dead mutant cofilin mimic SD? The lack of effect of the cofilin^S3D^ on control spines, synaptic strength and L&M (on short lasting LTP?) should be addressed.

2) The effect of 5 hours of sleep deprivation on spine density is surprisingly strong; in fact with such strong structural changes one would expect much greater behavioral deficits than just impaired learning. A well-known caveat of sleep deprivation experiment is the stress induced by the deprivation procedure. While the authors discussed that stress has the opposite effect on spine density, could the reduced spine density be caused by other factors associated with the deprivation procedure rather than sleep deprivation per se?

3) The authors show that the effect of sleep deprivation is restricted to CA1 region. This raises two questions: (i) why is CA1 specifically vulnerable to sleep deprivation, while other brain regions are not? What causes cofilin activity to be sensitive to sleep deprivation in hippocampus but not other brain regions? (ii) Can the effect of sleep deprivation on memory, measured at the behavioral level, be completely attributed to deficit in CA1 plasticity? Is there another behavior that measures non-CA1 plasticity dependent learning that is spared by sleep deprivation?

4) The anatomical specificity of the structural SD-induced changes (not involving CA3) and p-cofilin/cofilin SD-decrease (not seen in PFC) is interesting, raising the possibility that SD changes in CA1 is not a good model for the role of sleep in neocortex. If so (and the structural changes need assessment in the PFC), then the assertion that "Our data contradict the synaptic homeostasis hypothesis[…]", should be modified to reflect the anatomical specificity noted above.

5) Reducing cofilin activity seems to improve learning without any negative effect on behavior. So why hasn't cofilin activity been eliminated during evolution?

[Editors' note: further revisions were requested prior to acceptance, as described below.]

Thank you for resubmitting your work entitled "Sleep deprivation causes memory deficits by negatively impacting neuronal connectivity in hippocampal area CA1" for further consideration at *eLife*. Your revised article has been favorably evaluated by a Senior editor, a Reviewing editor, and two reviewers.

The authors have been very responsive including the provision of new experiments to help resolve some of the issues raised in the first round. Notably, they examined the effect of hippocampal application of an AAV with an activated form of cofilin3SA and found it sufficient to cause deficits in the memory task but it was without effect on the form of LLTP that was disrupted by sleep deprivation (SD; the spaced 4-train LTP). This dissociation of cofilin effect from SD effect suggests that cofilin activation, while likely to be necessary, is not sufficient to underlie SD induced changes in synaptic plasticity. Further, it cannot be ruled out that the disruption of memory by cofilin activation is mediated by the same mechanism as by SD (for example SD does disrupt the one form of LLTP- is this sufficient to disrupt memory?).

On the other hand, the apparent necessity of an intact cAMP-PDE4-PKA-LIMK-cofilin activation-signaling pathway for SD induced memory disruption and reduction of spine density is a significant set of observations of general interest and potentially high impact.

Specific recommendations:

1) Last sentence of the Abstract should be altered accordingly.

2) The same for the second to last sentence of the Introduction.

3) Is there a significant effect of cofilin^S3D^ in NSD vs. wildtype NSD on spine density?

4) In the second paragraph of the Discussion, it might be noted that the "one form" of LTP not impacted by active cofilin is the same as that disrupted by SD.

5) Is there a relevant phosphatase responsible for the SD induced change in pCofilin/cofilin ratio? Its activity is certainly implied by the findings of this study.

6) In the seventh paragraph of the Discussion, it might be noted that the hippocampal specific plasticity may also be mediated by different forms of plasticity whether or not they are more or less prominent.

---

## [Author Response]

*Essential revisions:*

*1) The causal association of SD effects on PDE and synaptic structure, LLTP and long-term memory are somewhat unclear. The experimental protocol involves training, immediately followed by 5h SD and a probe, 24h after that. This implies that LLTP disruption occurs at least an hour or more after induction of synaptic potentiation or that synaptic potentiation in CA1 normally occurs following the training, during sleep. Thus SD following training might disrupt the putative sleep potentiation as proposed. If the latter is occurring, then synaptic strength, dendritic structure and spine number should increase accordingly from the time immediately following training to 5h following training.*

*Alternatively, reduction of cAMP, PKA or cofilin inactivation should reduce LLTP an hour or more after LTP induction.*

*Another possibility is that the cAMP/PKA/cofilin SD associated changes and spine/dendritic structure, LLTP and/or memory changes are not mechanistically linked. Evidence for such linkage would seem essential to the proposed mechanisms of SD induced memory deficits.*

The reviewer describes two possibilities by which sleep deprivation mediates its effect on cognitive processes. The first possibility is that sleep deprivation disrupts learning-induced putative potentiation during sleep which may be reflected in increased cAMP/PKA signaling, cofilin inactivation, synaptic potentiation, and the formation of new spines. The alternative possibility is that sleep deprivation leads to cognitive deficits through other mechanisms. Our work focuses on the molecular mechanisms by which sleep deprivation negatively impacts synaptic plasticity and memory. We show that sleep deprivation causes spine loss in CA1 neurons of the hippocampus that is accompanied by reduced cofilin phosphorylation levels, a protein that regulates dendritic spine formation. We find that expression of a dominant negative inactive version of cofilin is sufficient to prevent the spine loss, synaptic plasticity deficits, and memory impairments associated with sleep deprivation. In addition, we show that the changes in cofilin signaling are directly mediated by the phosphodiesterase isoform PDE4A5 that inhibits the PKA/LIMK pathway. Therefore, we have shown that changes in cAMP/PKA/LIMK/cofilin signaling with sleep deprivation are sufficient to cause memory deficits, and our work using genetic manipulations of both ends of the cAMP/PKA/LIMK/cofilin pathway causally links changes in hippocampal PKA/LIMK/cofilin signaling with the spine loss, synaptic plasticity, and memory deficits caused by sleep deprivation.

Other researchers (Yang et al., 2014, Science) have explored changes in dendritic spines after learning. They have found that there are more spines on neurons activated after sleep. This result is consistent with the idea that sleep deprivation disrupts learning-induced changes in spines. These researchers have not manipulated spine formation, or cofilin signaling in their experiments so they have not examined any functional roles for spine formation during sleep. We focused on functional experiments to manipulate signaling pathways during sleep deprivation. We have not examined directly changes in spines with sleep as this would require in vivo 2 photon imaging in the hippocampus with behavior, experiments that go well beyond the scope of the current manuscript.

We now further clarify the relationship between our work with sleep deprivation and sleep in the manuscript. In the manuscript, we now included the following text in the manuscript in the Discussion: “In line with our finding of reductions in spines during sleep deprivation, recent work by Yang and colleagues revealed that sleep promotes dendritic spine formation in neurons activated by learning (Yang et al., 2014 Science). Combined with our work, these experiments suggest that sleep deprivation disrupts learning-induced changes in spines that occur during sleep. Importantly, our structural studies reveal that spine loss is reversed by recovery sleep, consistent with this idea.”

The other issue that is raised in this comment is the question of the relationship between synaptic plasticity (LTP) and memory. Our work shows that these “go together” that is, when one is reduced the other is also, but our work does not directly speak to the relationship between synaptic plasticity and memory. Our manipulations show that sleep deprivation leads to changes in cAMP/PKA/LIMK/cofilin and spines and that when we reverse these molecular and structural changes we block the effects of sleep deprivation on both synaptic plasticity and memory.

We now clarify these points about the relationship between molecular changes, synaptic plasticity and memory in the manuscript. In the Discussion section we now write: “In light of the fact that elevated cofilin activity can lead to spine shrinkage and spine loss (Zhou, Homma and Poo, 2004; Pontrello et al., 2012), our genetic manipulations of cofilin and PDE4A5 signaling independently links impairments in synaptic plasticity and memory caused by brief sleep deprivation with the loss of dendritic spines in the hippocampus. […] That care should be taken with directly relating LTP deficits with memory impairments is emphasized by our findings that expression of constitutively active cofilin is sufficient to cause memory deficits while it does not impact at least one form of L-LTP.”

*The primary conclusion drawn from this study is that 5h of SD reduced CA1 spine density and LLTP of the Schaffer Collateral synapses in association with hippocampal dependent learning and memory dysfunction, by increasing cofilin activity. The evidence presented includes 1) western blots showing 5h SD decreases the ratio of the pCofilin-inactive/cofilin-active form and 2) over-expression of an inactive mutant cofilin^S3D^ in CA1 prevented SD effects on spines, LLTP and L&M.*

*These data are preliminary in the sense that there is no evidence that cofilin activity was affected (or to what degree) by the cofilin^S3D^, there is no direct measure of the putative SD effect on cofilin and it is not clear how great an effect on cofilin activity is needed (i.e. is sufficient) to elicit the observed SD effect on spines, LLTP and L&M. Can off-target effects of the mutant be ruled out? Does a phospho-dead mutant cofilin mimic SD? The lack of effect of the cofilin^S3D^ on control spines, synaptic strength and L&M (on short lasting LTP?) should be addressed.*

Our work shows that sleep deprivation negatively impacts spine numbers in the hippocampus that is accompanied by increased activity of cofilin. To directly test the role of altered cofilin signaling in the spine loss associated with sleep deprivation, we expressed a mutant inactive version of cofilin which is known to inhibit the function of endogenous cofilin function (Zhao et al., 2008 Journal of Biological Chemistry) and prevents dendritic spine changes associated caused by increased endogenous cofilin activity (Shi et al., 2009Journal of Neuroscience). We analyzed spine density as a final common pathway of cofilin activity. We found that expression of the mutant cofilin prevents the spine loss caused by sleep deprivation. Furthermore, expression of the inactive form of cofilin also prevented the deficits in L-LTP and long-term memory formation. Thus, this work shows the necessity of increased cofilin activity for the expression of synaptic plasticity deficits and memory impairments associated with sleep deprivation and conceptually addresses how sleep deprivation leads to spine loss.

The possibility of off-target effects is unlikely as expression of the mutant prevents spine loss, LTP deficits, and memory impairments, bringing these measures back to “normal” levels. Further, cofilin^S3D^ expression did not impact memory in non-sleep deprived mice. This lack of an effect in non-sleep deprived mice shows the selectivity of our cofilin manipulation to the sleep-deprived state. This suggests that cofilin^S3D^ mutant is not having “off target” effects on other molecular processes.

We have now incorporated this point in the Results section: “Although we cannot rule out the possibility of off-target effects of the cofilinS3D mutant, we think that these are unlikely as expression of this mutant form of cofilin reversed the effects of sleep deprivation, restoring spine loss, LTP and memory to non-sleep deprived levels while not having an effect in non-sleep deprived mice.”

We agree with the reviewer that it would be interesting to determine whether expression of a phospho-dead mutant cofilin in hippocampal neurons is sufficient to mimic the behavioral and electrophysiological phenotypes associated with sleep deprivation. We now show that expression of cofilin^S3A^ is sufficient to cause cognitive deficits in the object-location memory task (Figure 4—figure supplement 2). We also show that spaced 4-train LTP was not affected by this manipulation. These new findings are now described in the Results section, subsection “Suppressing cofilin function in hippocampal neurons prevents the impairments in memory and synaptic plasticity caused by brief periods of sleep deprivation”. Together, these data suggest that at least at the behavioral level expression of constitutively active cofilin mimics the phenotypes associated with brief sleep deprivation and that care should be taken with directly relating LTP deficits with memory impairments.

These data are now also discussed in the Discussion section. Specifically, we write “In light of the fact that elevated cofilin activity can lead to spine shrinkage and spine loss ((Zhou, Homma and Poo, 2004; Pontrello et al., 2012), our genetic manipulations of cofilin and PDE4A5 signaling independently links impairments in synaptic plasticity and memory caused by brief sleep deprivation with the loss of dendritic spines in the hippocampus. […] That care should be taken to directly relate LTP deficits with memory impairments is emphasized by our findings that expression of constitutively active cofilin is sufficient to cause memory deficits while it does not impact at least one form of L-LTP.”

In future studies we will further assess the impact if cofilin^S3A^ expression on various other LTP protocols both in vivoand ex vivobut feel that these additional experiments are better suited for a separate manuscript. Such experiments address the question whether expression of constitutively active cofilin is sufficient to impair various forms of synaptic plasticity while the current manuscript focuses on the necessity of cofilin activity for the behavioral and electrophysiological deficits associated with sleep deprivation.

We would also like to emphasize that suppression of PDE4A5 function through overexpression of a catalytically inactive mutant version of PDE4A5 (PDE4A5catnul) restores levels of phospho-cofilin (Figure 5) and that this manipulation prevents the negative impact of sleep deprivation on memory consolidation (Figure 5). Thus, by manipulating cofilin phosphorylation through two entirely different signalling routes / inputs each served to prevent the memory deficits associated with sleep deprivation.

We have emphasized this important finding in the first paragraph of the Discussion: “Sixth, we find that suppression of localized PDE4A5 functioning through overexpression of a catalytically inactive mutant version of PDE4A5 to displace anchored, active endogenous PDE4A5 serves to change LIMK and cofilin phosphorylation levels associated with sleep deprivation. Finally, we show that hippocampal suppression of endogenous PDE4A5 function prevents the negative impact of sleep deprivation on memory consolidation. Thus, two different ways of manipulating cofilin phosphorylation prevent the memory deficits associated with sleep deprivation.”

*2) The effect of 5 hours of sleep deprivation on spine density is surprisingly strong; in fact with such strong structural changes one would expect much greater behavioral deficits than just impaired learning. A well-known caveat of sleep deprivation experiment is the stress induced by the deprivation procedure. While the authors discussed that stress has the opposite effect on spine density, could the reduced spine density be caused by other factors associated with the deprivation procedure rather than sleep deprivation per se?*

The reviewer raises the question whether, aside from stress, the procedure used to keep animals awake could lead to the spine loss associated with sleep deprivation. In one of our previous studies, we addressed this question by assessing whether stimulation itself negatively impacts memory. This does not appear to be the case. Specifically, we showed that subjecting the animals to the same amount of stimulation in the waking phase does not lead to memory impairments (Hagewoud et al., 2010 Sleep).

We now included the following sentence to the Discussion: “It is also unlikely that other factors associated with the procedure to keep animals awake rather than sleep deprivation per se causes the spine loss as our previous work indicated that applying the exact same amount of stimulation in the waking phase (i.e., the dark phase) does not lead to memory impairments (Hagewoud et al., 2010).”

3) The authors show that the effect of sleep deprivation is restricted to CA1 region. This raises two questions: (i) why is CA1 specifically vulnerable to sleep deprivation, while other brain regions are not? What causes cofilin activity to be sensitive to sleep deprivation in hippocampus but not other brain regions?

Our data indeed suggests that the hippocampus is specifically sensitive to short periods of sleep deprivation. For example, sleep deprivation leads to increased cofilin activity in the hippocampus, but not in the prefrontal cortex. We previously reported that the amygdala is less susceptible to sleep deprivation (Graves et al., 2003 Learning & Memory). Within the hippocampus, we have previously showed that five hours of sleep deprivation attenuates CREB phosphorylation in area CA1, but not area CA3 of the hippocampus (Vecsey et al., 2009 Nature). One reason why area CA1 may be more sensitive to sleep deprivation may be the high level of PDE4A5 expression in this specific hippocampal subregion (McPhee et al., 2001 Cellular Signalling).

In the manuscript, we have now included the following statement in the Discussion section: “Our work reveals that PDE4A5 is a critical mediator of the impact of sleep deprivation on memory consolidation, and one reason why hippocampal area CA1 may be specifically vulnerable to sleep deprivation may be the high level of PDE4A5 expression in this region (Shors, Chua and Falduto, 2001).”

*(ii) Can the effect of sleep deprivation on memory, measured at the behavioral level, be completely attributed to deficit in CA1 plasticity? Is there another behavior that measures non-CA1 plasticity dependent learning that is spared by sleep deprivation?*

There are two components to this comment. One is the relationship between plasticity and behavior, an issue that we address above in response to comment 1. The second is the about another behavior that might measure non-CA1 plasticity dependent learning that is spared by sleep deprivation. The current manuscript focuses on the impact of sleep deprivation on structural plasticity in area CA1 for reasons discussed above. We show that sleep deprivation reduces spine numbers in CA1 but not CA3. It will be interesting in future experiments to look at the impact of sleep deprivation on pattern completion and pattern separation, behavioral tasks that rely on hippocampal area CA3 and the dentate gyrus respectively (Nakazawa et al., 2002 Science; McHugh et al., 2007 Science) and at synaptic plasticity in these other hippocampal regions. These tasks require multiple training trials so direct examination of the impact of sleep deprivation selectively on memory consolidation is difficult. We have focused our work on sleep deprivation on single trial tasks so that we can look selectively at the impact of sleep deprivation on memory consolidation.

We have now added the following text to the Discussion to address this issue: “In the current study, we have focused on hippocampal area CA1 because we have previously found deficits in synaptic plasticity in this region caused by sleep deprivation (Vecsey et al., 2009 Nature). We found that sleep deprivation affects spine numbers in area CA1, but not area CA3. It will be interesting in future experiments to look at the impact of sleep deprivation on pattern completion and pattern separation, behavioral tasks that rely on hippocampal area CA3 and the dentate gyrus respectively (Nakazawa et al., 2002 Science; McHugh et al., 2007 Science) and at synaptic plasticity in these other hippocampal regions. Further, future experiments can restrict manipulations of cofilin function to area CA1 using viral constructs that are Cre-dependent in combination with Cre lines that express selectively in CA1, CA3 or the dentate gyrus.”

*4) The anatomical specificity of the structural SD-induced changes (not involving CA3) and p-cofilin/cofilin SD-decrease (not seen in PFC) is interesting, raising the possibility that SD changes in CA1 is not a good model for the role of sleep in neocortex. If so (and the structural changes need assessment in the PFC), then the assertion that "Our data contradict the synaptic homeostasis hypothesis….", should be modified to reflect the anatomical specificity noted above.*

We agree with the reviewer that there may be brain region specificity in the effects of sleep deprivation, as our own study shows at the level of cofilin phosphorylation. However, we would like to point out that it has been suggested that the synaptic homeostasis hypothesis not only applies to the cortex but to the hippocampus as well (Vyazovskiy et al., 2008 Nature Neuroscience). As such we feel it is important to emphasize that our data are not in line with the original proposition.

We have now included the following sentence in the Discussion section: “This hypothesis has focused on explaining data from the cortex rather than the hippocampus, but one previously published study has suggested that the synaptic homeostasis hypothesis applies to the hippocampus as well (Vyazovskiy et al., 2008). However, the hippocampus may be a special case from the cortex as the hippocampus is involved in episodic memory and in much greater experience dependent plasticity than anywhere else in the brain and thus our findings may not extend to other areas where synaptic plasticity is not as prominent.”

*5) Reducing cofilin activity seems to improve learning without any negative effect on behavior. So why hasn't cofilin activity been eliminated during evolution?*

This is an interesting point raised by the reviewer. Cofilin has a variety of functions in the body and, in this regard, the fact that cofilin has not been ‘deleted’ from the CNS indicates that it has evolutionary important functions that outweigh any negative effects on learning. Indeed, knockout of the cofilin-1 gene in mice is embryonic lethal (Gurniak et al., 2004 Developmental Biology). Furthermore, it is also important to note that structural plasticity is highly dynamic and for this process to be highly dynamic both positive and negative regulators of spine plasticity are required. Cofilin could for example play an essential role updating outdated memories, erasing irrelevant information, extinction learning, and transferring information from the hippocampus to cortical layers for long-term storage (i.e.systems consolidation). However, to date no studies have defined a role for cofilin in these processes.

We have now included the following statement in the Introduction section of the manuscript “In addition to a critical function during development (Gurniak, Perlas and Witke, 2005), cofilin plays an essential role in synapse structure by mediating both the enlargement and pruning of dendritic spines (Rust, 2015; Bamburn, 1999; Bosch et al., 2014).”

[Editors' note: further revisions were requested prior to acceptance, as described below.]

*Specific recommendations:*

*1) Last sentence of the Abstract should be altered accordingly.*

We agree with the reviewers and have now revised the last sentence of the Abstract. The last sentence of the abstract now reads “Our work demonstrates the necessity of an intact cAMP-PDE4-PKA-LIMK-cofilin activation-signaling pathway for sleep deprivation-induced memory disruption and reduction in hippocampal spine density”.

2) The same for the second to last sentence of the Introduction.

The second to last sentence of the Introduction has now been altered to “Thus changes in the cAMP-PDE4-PKA-LIMK-cofilin signaling pathway in the adult hippocampus underlie the cognitive deficits associated with sleep loss.”

*3) Is there a significant effect of cofilin^S3D^ in NSD vs. wildtype NSD on spine density?*

The reviewers raise an interesting point. Previous work by others has indicated that cofilin^S3D^ expression per se does not alter hippocampal spine density (Shi et al., 2009 Journal of Neuroscience; Pontrello et al., 2012 *PNAS*). In the Results section we now write “Importantly, cofilin^S3D^ expression does not alter spine density under baseline conditions (Pontrello et al., 2012; Shi et al., 2009).”

*4) In the second paragraph of the Discussion, it might be noted that the "one form" of LTP not impacted by active cofilin is the same as that disrupted by SD.*

We agreed with the reviewers and have revised the text accordingly. The sentence now reads “That care should be taken to directly relate LTP deficits with memory impairments is emphasized by our findings that expression of constitutively active cofilin is sufficient to cause memory deficits while it does not impact at least one form of L-LTP that is disrupted by sleep deprivation (Seabold et al., 2010).”

*5) Is there a relevant phosphatase responsible for the SD induced change in pCofilin/cofilin ratio? Its activity is certainly implied by the findings of this study.*

While the phosphatase slingshot could contribute to the altered pcofilin/cofilin ratio we would like to point out that the changes in PKA-LIMK signaling seem to play a causal role in the altered pcofilin/cofilin ratio, because blocking the degradation of cAMP by PDE4A5 is sufficient to fully restore the pcofilin/cofilin ratio in sleep deprived mice (Figure 5). Nevertheless, in the revised manuscript in the Results section we now include a sentence about a potential role for the phosphatase slingshot in the altered pcofilin/cofilin ratio observed in sleep deprivation mice.

We now write “While expression of PDE4A5^catnull^ fully restored the pcofilin/cofilin ratio in the hippocampus of sleep deprived mice to the levels observed under non-sleep deprivation conditions, it should be noted that phosphatases such as slingshot (Sarmiere and Bamburg, 2004) may also contribute to the reduction in cofilin phosphorylation levels under conditions of sleep deprivation.”

*6) In the seventh paragraph of the Discussion, it might be noted that the hippocampal specific plasticity may also be mediated by different forms of plasticity whether or not they are more or less prominent.*

We appreciate the suggestion and revised the text accordingly.

We now write “However, the hippocampus may be unique from the cortex as the hippocampus is involved in episodic memory and in much greater experience-dependent plasticity than anywhere else in the brain and thus our findings may not extend to other areas where synaptic plasticity is not as prominent. Further, the hippocampus also exhibits many distinct forms of synaptic plasticity.”